# Regulation of A-to-I RNA editing and stop codon recoding to control selenoprotein expression during skeletal myogenesis

Yuta Noda[1], Shunpei Okada[1,2] & Tsutomu Suzuki [1✉]

Selenoprotein N (SELENON), a selenocysteine (Sec)-containing protein with high reductive activity, maintains redox homeostasis, thereby contributing to skeletal muscle differentiation and function. Loss-of-function mutations in *SELENON* cause severe neuromuscular disorders. In the early-to-middle stage of myoblast differentiation, SELENON maintains redox homeostasis and modulates endoplasmic reticulum (ER) $Ca^{2+}$ concentration, resulting in a gradual reduction from the middle-to-late stages due to unknown mechanisms. The present study describes post-transcriptional mechanisms that regulate *SELENON* expression during myoblast differentiation. Part of an Alu element in the second intron of *SELENON* pre-mRNA is frequently exonized during splicing, resulting in an aberrant mRNA that is degraded by nonsense-mediated mRNA decay (NMD). In the middle stage of myoblast differentiation, ADAR1-mediated A-to-I RNA editing occurs in the U1 snRNA binding site at 5′ splice site, preventing Alu exonization and producing mature mRNA. In the middle-to-late stage of myoblast differentiation, the level of Sec-charged $tRNA^{Sec}$ decreases due to downregulation of essential recoding factors for Sec insertion, thereby generating a premature termination codon in *SELENON* mRNA, which is targeted by NMD.

[1] Department of Chemistry and Biotechnology, Graduate School of Engineering, University of Tokyo, 7-3-1 Hongo, Bunkyo-ku, Tokyo 113-8656, Japan.
[2] Department of Microbiology, Faculty of Medicine, Shimane University, 89-1 Enyacho, Izumo, Shimane 693-8501, Japan. ✉email: ts@chembio.t.u-tokyo.ac.jp

Skeletal muscle is the largest organ in the human body. It constitutes approximately 40% of total body weight and plays fundamental roles in exercise, postural maintenance, breathing, regulation of energy metabolism, and other processes. For muscle contraction and/or normal differentiation of skeletal muscle, precise control of intracellular calcium ion ($Ca^{2+}$) homeostasis is critical[1,2]. Muscle cells contain a specialized form of endoplasmic reticulum (ER) called sarcoplasmic reticulum (SR), which is dedicated to $Ca^{2+}$ storage and release for contraction and relaxation. Cytoplasmic $Ca^{2+}$ concentration is regulated mainly by the sarco-endoplasmic reticulum $Ca^{2+}$-ATPase (SERCA) pump, the inositol-1,4,5-triphosphate receptor ($IP_3R$), and the ryanodine receptor (RyR), all of which reside on the ER membrane. Aging or physical inactivity diminishes $Ca^{2+}$ release from the SR[3], leading to skeletal muscle atrophy, contractile impairment, and metabolic abnormalities[4,5].

Skeletal muscle consists of bundles of multinucleated muscle fibers that are constantly repaired and regenerated by stem cells known as muscle satellite cells (mSCs). Upon injury, mSCs become activated, proliferate, differentiate into myoblasts, and assemble to form multinucleated myotubes, ultimately fusing with muscle fibers. These processes are highly regulated by the expression of a series of myogenic regulatory factors, including MYOD1 and MYOG, microRNAs (e.g., miR-1, miR-206, and miR-133), and muscle structural proteins (e.g., myosin heavy chain)[6,7]. During myogenesis, massive levels of reactive oxygen species (ROS) are generated[8]. ROS play a critical role in promoting skeletal muscle differentiation[9–11]. ROS induces the p38 mitogen-activated protein kinase signaling pathway to antagonize the proliferation-promoting JNK pathway, thereby facilitating myogenesis[12]. However, high levels of ROS impair redox homeostasis and induce oxidative stress, leading to pathological consequences such as muscular dystrophies[13]. To maintain appropriate ROS levels and decrease their toxic effect on myogenesis, myoblasts use antioxidant enzymes, including selenoproteins.

Selenoproteins contain selenium (Se) in the polypeptide chain as a selenocysteine (Sec) residue. Sec is the 21st naturally occurring amino acid that is co-translationally incorporated into polypeptide chains. Sec is incorporated into selenoproteins at the in-frame UGA codon through a unique translation system called "recoding"[14]. Many *trans* factors and *cis* elements are required for specific insertion of Sec at the UGA recoding site[15–17] (Supplementary Fig. 1). Downstream of the recoding site in mRNA, a unique stem-loop structure called the selenocysteine insertion sequence (SECIS) works as a *cis* element for Sec insertion. A special tRNA for Sec, tRNA$^{Sec}$, is first serylated by seryl-tRNA synthetase (SARS1) and then phosphorylated to yield phosphoseryl(pSer)-tRNA$^{Sec}$ by phosphoseryl-tRNA kinase (PSTK). Selenophosphate, a selenium donor for Sec formation, is synthesized by selenophosphate synthetases (SEPHS1 and SEPHS2). Finally, *O*-phosphoseryl-tRNA$^{Sec}$ selenium transferase (SEPSECS) uses selenophosphate to convert pSer-tRNA$^{Sec}$ to Sec-tRNA$^{Sec}$. Sec-tRNA$^{Sec}$ is recognized specifically by a specialized elongation factor called EEFSEC, followed by interaction with the SECIS element with the help of SECISBP2; it is then recruited to the ribosome translating the recoding site. Insufficiencies in Sec-tRNA$^{Sec}$ and/or the Sec insertion machinery may result in the recognition of the UGA recoding site as a premature termination codon (PTC), which is then targeted by nonsense-mediated mRNA decay (NMD)[18]. In addition, translation terminates at the recoding site, generating a truncated isoform[19,20].

Selenoproteins have unique redox properties due to the high nucleophilicity of Se[21]. Se is an essential element, and plays major physiological roles in thyroid hormone metabolism, immunity, and antioxidant defenses[22,23]. Deficiencies in Se are associated with various pathologic conditions, including cardiovascular disease, cancer, hepatopathy, and arthropathy[24]. In addition, Se is required for the normal physiological state of muscles[25], and its deficiency can lead to acquired cardiomyopathy[26] or white muscle disease[27]. By contrast, Se is a toxic metal, and excessive Se intake causes acute selenosis[24].

The human genome encodes 25 selenoproteins, many of which are involved in antioxidant defenses and redox signaling by neutralizing ROS or repairing oxidized macromolecules[17,28]. To date, however, only a few selenoproteins have been functionally characterized. Among them, *selenoprotein N* (*SELENON*, also known as *SEPN1*) is associated with human genetic disorders[29,30]. Specifically, loss-of-function mutations in *SELENON* cause several neuromuscular disorders, collectively termed SELENON-related myopathies (SELENON-RM)[31]. These conditions include rigid spine muscular dystrophy (RSMD1)[32,33], the classical form of multiminicore disease (MmD)[34], desmin-related myopathy with Mallory-body-like inclusions (MB-DRM)[35], and congenital fiber-type disproportion (CFTD)[36]. Biopsies yield results typical of congenital myopathy, including minicores, type 1 fiber predominance, and mild endomysial fibrosis[31]. SELENON-RM is typically associated with respiratory insufficiency, scoliosis, and congenital muscle weakness, especially in axial muscles, which are particularly vulnerable to oxidative stress. However, the molecular mechanisms underlying these diseases remain largely unknown, and no practical cures are available.

SELENON is a glycoprotein bearing a Sec residue at its active site, and is localized in the ER/SR membrane[37]. SELENON plays two main roles. First, it maintains redox homeostasis by decreasing ROS levels to protect cells from oxidative stress. Consistent with this, primary cells from SELENON-RM patients are susceptible to hydrogen peroxide-induced oxidative stress[38]. Second, SELENON modulates $Ca^{2+}$ concentration in the ER. SERCA is a major $Ca^{2+}$-ATPase pump on the ER/SR that regulates ER $Ca^{2+}$ concentration. The activity of SERCA is tightly regulated by the redox status of a Cys residue in the L4 luminal domain of SERCA[39]. SELENON uses an EF-hand domain that senses the luminal $Ca^{2+}$ level[40]. Low levels of $Ca^{2+}$ in the ER induce a conformational change in SELENON and reduce the oxidized thiol group of the Cys residue to activate SERCA, thereby promoting $Ca^{2+}$ uptake[41–43]. *SELENON*-deficient myotubes exhibit elevated levels of cytoplasmic $Ca^{2+}$ and reduced levels of SR $Ca^{2+}$, resulting in abnormal $Ca^{2+}$ homeostasis[38]. Hence, SELENON is an essential protein for muscle regeneration and satellite cell maintenance[44]. In skeletal myogenesis, the steady-state level of SELENON is high in the early myoblast stage and decreases gradually by the time of myotube formation[37], suggesting that downregulation of *SELENON* is important for skeletal myogenesis. However, the underlying molecular mechanism and its physiological significance remain largely unknown.

This study identified post-transcriptional regulatory mechanisms of *SELENON* expression during myoblast-to-myotube differentiation. The Alu element, a short DNA segment of ~300 nucleotides (nt), is a primate-specific retrotransposon belonging to the short interspersed nucleotide elements (SINE) family[45]. *SELENON* precursor mRNA has an antisense strand Alu element bearing a cryptic splicing signal in the second intron. Part of this Alu element (Alu exon) is frequently exonized during splicing, yielding an aberrant transcript that is constitutively degraded by NMD. Intriguingly, the frequency of Alu exonization of *SELENON* mRNA is higher in skeletal muscle than in other tissues[46] (Supplementary Fig. 2). Therefore, we hypothesized that increased inclusion of the Alu element during myoblast differentiation would reduce steady-state levels of mature *SELENON* mRNA, and investigated the mechanism underlying the

regulation of Alu exonization during myoblast differentiation. We found that in the early stage of myoblast differentiation, hnRNP C plays a major role in preventing the Alu element from exonization. In the middle stage, A-to-I RNA editing occurs in the U1 snRNA binding site at the 5′ splice site (5′SS) of the Alu exon, mediated by ADAR1, preventing Alu exonization and allowing the production of mature mRNA. In the middle-to-late stage, both hnRNP C and ADAR1 are downregulated, and *SELENON* expression decreases due to elevated Alu exonization to generate an aberrant mRNA that is degraded by NMD. Furthermore, in the late stage, the level of Sec-charged tRNA$^{Sec}$ is reduced due to downregulation of SEPSECS and other recoding factors for Sec insertion, thereby suppressing UGA/Sec recoding. Consequently, *SELENON* mRNA is degraded by NMD, generating a short isoform of SELENON with no catalytic activity.

## Results

### ADAR1 and hnRNP C protect against aberrant Alu exonization of *SELENON* mRNA.

To investigate the mechanism underlying the regulation of *SELENON* gene expression during skeletal myogenesis, we first focused on aberrant exonization of the Alu element in *SELENON* mRNA. Like other human genes, *SELENON* also has several Alu elements in its intronic regions. The antisense Alu element has a cryptic splicing signal including a polyuridine tract in the region upstream of the 3′ splice site (3′SS). Consequently, once the antisense strand of the Alu element is transcribed, a part of the Alu element is frequently recognized by the splicing machinery and inserted as an Alu exon in the mature mRNAs[47]. *SELENON* has one sense Alu element (S-Alu) and three antisense Alu elements (AS-Alu1, 2, and 3) in its second intron (Fig. 1a, b). A part of AS-Alu1 (102 nt) is exonized during splicing to generate the Alu exon-containing transcript (inclusion isoform) (Fig. 1b, c)[33].

Using total RNAs from HeLa cells (Fig. 1d) and human tissues (Supplementary Fig. 2), we amplified the region spanning exons 2 and 3 of *SELENON* mRNA by RT-PCR, and confirmed the presence of the inclusion isoform with the Alu exon, although the mature mRNA (skipping isoform) was predominant. Of all human tissues, skeletal muscle shows the highest frequency of the skipping isoform (Supplementary Fig. 2). According to our previous study[48], A-to-I RNA editing mediated by ADAR1 antagonizes Alu exonization in human culture cells. Hence, we knocked down either *ADAR1* or *ADAR2* in HeLa cells (Supplementary Fig. 3) and quantified the inclusion isoform by RT-qPCR (Fig. 1d). The inclusion isoform was clearly upregulated when *ADAR1* was knocked down but not when *ADAR2* was depleted, indicating that ADAR1 suppresses the Alu exon in *SELENON* mRNA.

Next, we considered whether hnRNP C is a potential regulator of Alu exonization, as hnRNP C suppresses the Alu exon by competing with U2AF65 in a transcriptome-wide manner[49,50]. In addition, the iCLIP signal of hnRNP C resides in AS-Alu1 in the *SELENON* gene (Fig. 1b)[49]. When we knocked down hnRNP C (Supplementary Fig. 3), the level of the inclusion isoform increased as expected (Fig. 1d), indicating that hnRNP C is another regulatory factor that prevents Alu exonization in *SELENON* mRNA.

Because the inclusion isoform containing Alu exon has a PTC (Fig. 1c), we assessed whether the inclusion isoform is targeted by NMD. Knockdown of *UPF1* (Supplementary Fig. 3), which encodes a factor essential for NMD resulted in the accumulation of the inclusion isoform (Fig. 1d), demonstrating that the inclusion isoform of *SELENON* mRNA is constitutively degraded by NMD. This finding may explain our observation of the limited upregulation of the inclusion isoform when either *ADAR1* or

*hnRNP C* was knocked down. Double knockdown of *ADAR1* and *UPF1* significantly increased the amount of the inclusion isoform from 4.6 to 14% (Fig. 1e). In addition, knockdown of both *hnRNP C* and *UPF1* increased the amount of the inclusion isoform from 6.2% to 19.3% (Fig. 1e). These data confirmed that the inclusion isoform is constantly degraded by NMD, and that the level of the skipping isoform is significantly downregulated. Because ADAR1 and hnRNP C redundantly block Alu exonization, the level of the inclusion isoform would likely be much higher when both *ADAR1* and *hnRNP C* are silenced simultaneously with *UPF1* knockdown.

The levels of expression of *ADAR1* and *hnRNP C* were investigated in multiple tissues using human data from the Genotype-Tissue Expression (GTEx) project[51]. The level of *ADAR1* was lower in skeletal muscle than in all tissues (Supplementary Fig. 4a). The overall editing level was also found to be much lower in skeletal muscle than in other tissues[52]. Moreover, the level of *hnRNP C* was also relatively low in skeletal muscle (Supplementary Fig. 4b). These findings explain the high levels of accumulation of the Alu exon of *SELENON* mRNA in mature skeletal muscle (Supplementary Fig. 2).

### A-to-I editing at the 5′SS antagonizes Alu exonization of *SELENON* mRNA.

ADAR1 is an RNA-modifying enzyme responsible for A-to-I RNA editing specific to double-stranded RNA (dsRNA) regions[53]. The vast majority of A-to-I RNA editing sites in the human transcriptome are present in dsRNA regions formed by inverted Alu elements[54,55]. The inosine chemical erasing (ICE) method[48,56] was utilized to evaluate A-to-I RNA editing sites in the Alu elements in the second intron of *SELENON* pre-mRNA in the human brain and testis. The ICE method identified 36, 24, 5, and nine sites in S-Alu, AS-Alu1, AS-Alu2, and AS-Alu3, respectively (Fig. 1b, c, Supplementary Fig. 5, and Supplementary Data 1). Like other A-to-I editing sites in Alu elements[54], most of these sites had a low (<20%) editing frequency (Supplementary Data 1). Because A-to-I editing occurs in long double-stranded regions, it is likely that S-Alu forms a duplex structure with AS-Alu1, AS-Alu2, or AS-Alu3 (Fig. 1b). Judging from the density and frequency of A-to-I editing, S-Alu preferentially forms a duplex with AS-Alu1. We then examined whether ADAR1 is responsible for A-to-I editing in AS-Alu1. The editing frequencies of three sites (Nos. 46, 57, and 60) in AS-Alu1 decreased upon ADAR1 knockdown, but did not change upon ADAR2 knockdown (Supplementary Fig. 3, 6), demonstrating that these sites are actually edited by ADAR1.

The A-to-I editing sites were mapped onto the putative secondary structure formed by S-Alu and AS-Alu1 (Supplementary Fig. 7). The Alu exon in the inclusion isoform contains six editing sites (Nos. 39–44) (Fig. 1c). The editing frequency of these six sites was markedly lower in the inclusion isoform of *SELENON* than in the precursor mRNA (Supplementary Fig. 8), indicating that the pre-mRNA bearing a highly edited AS-Alu1 segment to be eliminated by the splicing machinery, allowing the non-edited Alu exon to accumulate in the inclusion isoform. Next, we focused on two editing sites (Nos. 45 and 46) in the GURAG motif at 5′SS of the Alu exon (Fig. 1c, f) because they are within the sequence directly recognized by U1 snRNA (Fig. 1f). The editing frequency of site No. 46 was much higher than that of site No. 45 (Fig. 1f). These two adenosines in the 5′SS pair with pseudouridines (Ψ) in U1 snRNA (Fig. 1f). We speculated that once they are edited, I-Ψ base-pairs might destabilize recognition of the 5′SS by U1 snRNA (Fig. 1f), thereby impairing Alu exonization. To investigate this hypothesis, we made a minigene reporter construct (WT) bearing the Alu exon together with exons 2 and 3 of *SELENON* pre-mRNA (Fig. 1g). To mimic A-to-I editing at 5′SS, we introduced an A-to-G mutation at site No. 46

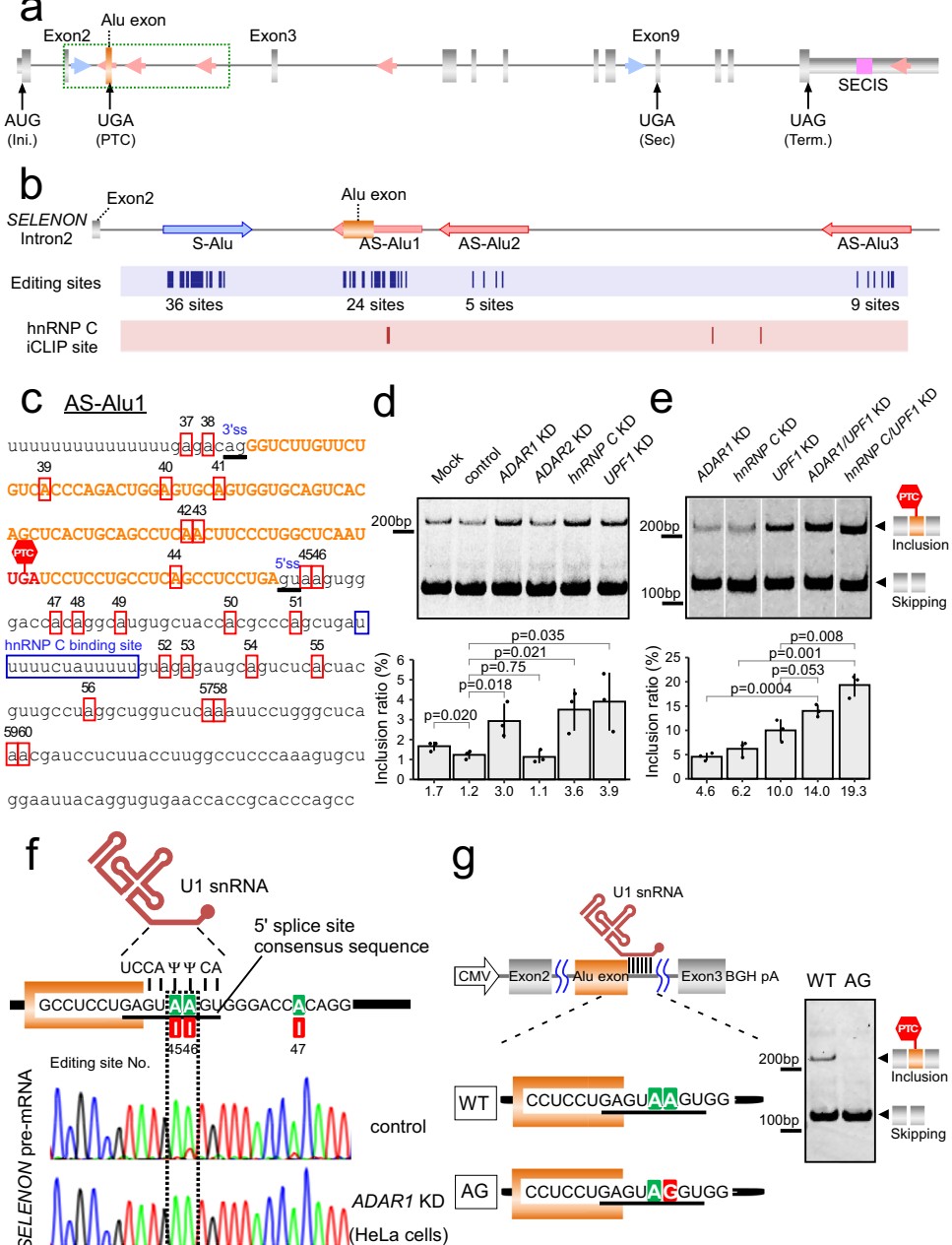

**Fig. 1 ADAR1 and hnRNP C suppress Alu exonization of *SELENON* mRNA. a** Human *SELENON* gene with exons (boxes) and introns (lines). The Alu exon and SECIS are indicated by orange and magenta boxes, respectively. Sense and antisense strands of Alu elements are indicated by blue and red arrows, respectively. Int. and Term. represent initiation and termination codons. **b** Intronic region of *SELENON* including four Alu elements (green dashed box in **a**). Purple column represents A-to-I editing sites determined by ICE method. Pink column represents iCLIP sites for hnRNP C[49]. **c** RNA sequence of AS-Alu1 in *SELENON*. A-to-I editing sites are boxed in red, with site numbers shown in Supplementary Fig. 5. hnRNP C binding site identified by iCLIP is boxed. The Alu exon sequence is colored in orange. 3′SS and 5′SS of the Alu exon are indicated by underbars. PTC (UGA) is indicated in red. **d** Alu exonization of *SELENON* in HeLa cells upon knockdown of each target mRNA. Control siRNA targeted luciferase. Upper panel shows RT-PCR products of a part of *SELENON* mRNA spanning exons 2 and 3. Lower panel shows the inclusion ratio (%), determined by RT-qPCR. Data are presented as mean values ±S.D. Statistical significance was determined by an unpaired two-tailed *t* test; *n* = 3 biologically independent samples. **e** Alu exonization of *SELENON* in HeLa cells upon double-knockdown of target mRNAs. Upper panel shows the inclusion and skipping isoforms with a schematical illustration. Lower panel shows the inclusion ratio (%), as described in **d**, above. **f** cDNA sequences of the Alu exon 5′SS in *SELENON* pre-mRNAs from HeLa cells treated with siRNAs targeting luciferase (control) and *ADAR1* (KD). The editing sites are indicated. Consensus sequence of 5′SS (underlined) base-pairs with a part of U1 snRNA. The electropherograms for A, G, T, and C are colored green, red, gray, and blue, respectively. Ψ, pseudouridine. **g** Minigene constructs bearing the Alu exon (WT) or its editing-mimic mutant (AG). Skipping and inclusion isoforms from the minigene were detected by RT-PCR. This experiment was done once. Source data and unprocessed gel images are provided in Source Data file.

in the minigene construct (AG) (Fig. 1g). These constructs were introduced into HeLa cells. The inclusion isoform containing the Alu exon was clearly observed in the WT construct but not in the AG construct (Fig. 1g). These data strongly suggest that A-to-I editing at 5′SS plays a role in suppressing Alu exonization. To our knowledge, this is the first example of mRNA splicing regulated by A-to-I RNA editing at the U1 snRNA recognition site.

**Suppression of the Alu exon maintains *SELENON* mRNA integrity during myoblast differentiation.** Next, we investigated how *SELENON* expression is regulated during myoblast differentiation. To this end, we cultured human myogenic cells (Hu5/KD3) and induced in vitro differentiation from myoblasts to myotubes[57,58] (Fig. 2a, b). Upon induction, Hu5/KD3 cells become elongated, fuse to form multinuclear cells, and differentiate into myotubes 5 days after induction (Fig. 2b). The cells were harvested on Days 0, 1, 2, 3, 4, and 5 of induction and then subjected to western blotting (Fig. 2c) and transcriptome-wide analysis using RNA-seq (Fig. 2d, e and Supplementary Fig. 9). As differentiation progressed, we observed global changes in the transcriptome (Fig. 2d and Supplementary Fig. 9). Based on the expression levels of MYOD1, MYOG, and myosin heavy chain (MYH) (Fig. 2c, e)[59], we defined three differentiation stages: early stage (Day 0 and 1), middle stage (Day 2 and 3), and late-stage (Day 4 and 5) (Fig. 2a, c). Gene ontology enrichment analysis of the RNA-seq data revealed characteristic features of muscle differentiation (Supplementary Fig. 10). As previously reported[37], the steady-state level of SELENON protein decreased during the late stage of myoblast differentiation (Fig. 2c). In addition, we noted the presence of a short isoform of SELENON in the middle and late stages (Fig. 2c). Downregulation of *SELENON* mRNA during myoblast differentiation was confirmed by RNA-seq (Fig. 2e) and RT-qPCR (Supplementary Fig. 11). The steady-state levels of precursor and mature mRNAs of *SELENON* were quantified separately. Compared to the early stage (Day 0), the level of pre-mRNA was reduced to about 60% during the middle and late stages (Supplementary Fig. 11), suggesting that transcriptional regulation may be involved in *SELENON* expression. Compared with the pre-mRNA level, mature mRNAs rapidly decreased during the middle and late stages (Supplementary Fig. 11), indicating that *SELENON* pre-mRNA is also regulated by post-transcriptional processes. As differentiation progressed, hnRNP C and ADAR1 were downregulated at both the protein (Fig. 2c) and RNA levels (Fig. 2e), as reported previously[60,61], indicating that Alu exonization in *SELENON* mRNA is not suppressed by these factors during the late stage of myoblast differentiation. We then measured the inclusion and skipping isoforms of *SELENON* mRNA by RT-qPCR. As expected, the inclusion isoform gradually and significantly increased during the middle and late stages of differentiation (Fig. 2f). Given that the inclusion isoform is constantly degraded by NMD, we concluded that Alu exonization in the late stage was underestimated.

**Dynamic alteration of A-to-I RNA editing during myoblast differentiation.** Next, we analyzed A-to-I editing of *SELENON* pre-mRNA during myoblast differentiation and observed dynamic alteration of A-to-I editing in the AS-Alu1 region containing the Alu exon (Fig. 3a and Supplementary Data 1). When we compared the editing frequency of AS-Alu1 between Day 0 and 3 (Fig. 3a), we found that most of the 24 sites in this region were edited more frequently on Day 3 (Fig. 3a, b and Supplementary Fig. 12). At site No. 46, ~10% of *SELENON* mRNA was edited on Day 0 (Fig. 3c, d). On Day 3, the level of editing increased, reaching a maximum of 40% (Fig. 3c, d). Subsequently, it decreased rapidly, reaching 10% on Day 5 (Fig. 3c, d). On Day

0, both hnRNP C and ADAR1 were highly expressed (Fig. 2c, e), but the editing frequency at site No. 46 remained low (Fig. 3c, d). Because in the early stage, hnRNP C tightly binds AS-Alu1 and suppresses the Alu exon[49,50] (Fig. 3e), AS-Alu1 does not form a duplex with the S-Alu exon. In the middle stage, the level of hnRNP C decreased dramatically, relieving AS-Alu1 and allowing it to form a duplex with S-Alu, which serves as a substrate for ADAR1 (Fig. 3e). Thus, AS-Alu1 becomes highly edited on Day 3 (Fig. 3a, c, d and Supplementary Fig. 12), and the editing frequency at site No. 46 increases to prevent U1 snRNA recognition (Fig. 3e). From the middle-to-late stage, the editing rate decreases rapidly (Fig. 3c, d) as the steady-state level of ADAR1 gradually decreases (Fig. 2c, e). These observations suggest that dynamic alteration of A-to-I editing of *SELENON* pre-mRNA is achieved by a large conformational change of the intron 2 region in *SELENON* pre-mRNA, mediated by rapid depletion of hnRNP C in the middle stage (Fig. 3e). Supporting this finding, we observed clear promotion of A-to-I editing in the AS-Alu1 region in HeLa cells upon knockdown of hnRNP C (Supplementary Fig. 13 and Supplementary Data 1).

**Recoding regulation suppresses *SELENON* expression at the late stage of myoblast differentiation.** In the middle-to-late stages, the short isoform of SELENON appeared concomitantly with a dramatic decrease in the level of full-length SELENON (Fig. 2c). Given that the antibody's epitope resides in the upstream region of the UGA recoding site of SELENON (Supplementary Fig. 14), we speculated that the short isoform is a truncated product of SELENON terminated at the UGA recoding site (Supplementary Fig. 14). To test this possibility, we cloned the skipping isoform of *SELENON* with an N-terminal FLAG tag and the SECIS element in the 3′-UTR in order to express mature SELENON with Sec insertion (Fig. 4a). Based on this WT construct, we generated two mutant constructs, one lacking the SECIS element for expression of the truncated isoform terminating at the recoding site (ΔSECIS construct) (Fig. 4a), and the other in which the UGA recoding site was replaced with the GGA codon to express a full-length isoform without Sec insertion (GGA construct) (Fig. 4a). These constructs were introduced into HEK293T cells, which were then subjected to western blotting with an anti-FLAG antibody. Two bands corresponding to the full-length and short isoforms were observed in cells expressing the WT construct (Fig. 4b): one band corresponding to the truncated product in cells expressing the ΔSECIS construct, and one band corresponding to the full-length isoform in cells expressing the GGA construct. The short isoform has the same size as the truncated product of the ΔSECIS construct (Fig. 4b), suggesting that the short isoform of SELENON is derived from translation termination at the UGA recoding site.

The 25 selenoproteins in the human genome are divided into two classes based on the SECIS elements in the 3′-UTR of their mRNAs:[62,63] six selenoproteins have a type 1 SECIS element, whereas the others have a type 2 SECIS element, and the type 1 selenoproteins are susceptible to Se availability[62] (Supplementary Fig. 15a). One type 1 selenoprotein other than *SELENON*, *glutathione peroxidase 1* (*GPX1*), was detected in RNA-seq analysis of differentiating Hu5/KD3 cells (Supplementary Fig. 15a). Similar to *SELENON*, *GPX1* is also downregulated during myoblast differentiation at both the mRNA (Supplementary Fig. 15a) and protein (Supplementary Fig. 15b) levels. Based on the location of the UGA/Sec codon relative to the exon–exon junctions, *GPX1* mRNA should be degraded by NMD[18], due to the reduction of UGA/Sec recoding efficiency during the late stage of myoblast differentiation. These findings indicated that *SELENON* is downregulated by recoding regulation during the late stage of myoblast differentiation.

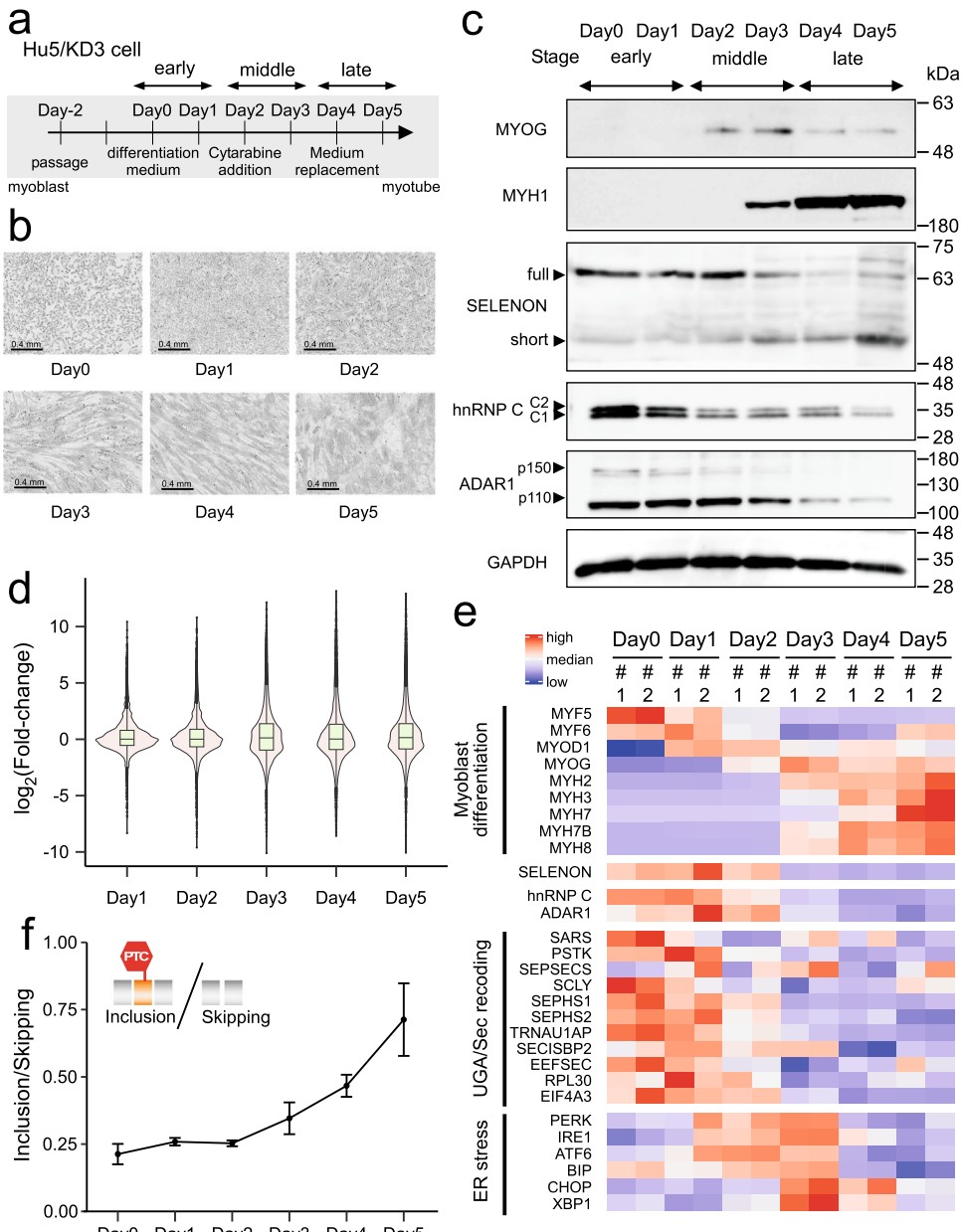

**Fig. 2 Stage-specific expression profiles and dynamic alteration of Alu exonization of *SELENON* during skeletal myogenesis of Hu5/KD3 cells.**
**a** Experimental schedule for differentiation of Hu5/KD3 cells. **b** Phase-contrast images showing Hu5/KD3 cell differentiation from Day 0 to 5. This experiment was repeated more than three times independently with similar results. **c** Expression analyses of the differentiation markers (MYOG, MYH1), SELENON, hnRNP C, ADAR1, and GAPDH (control) by western blotting on the indicated days during differentiation of Hu5/KD3 cells. SELENON (full-length and short isoforms), hnRNP C2 and hnRNP C1, and p150 and p110 isoforms of ADAR1, are indicated by arrowheads. The samples derived from the same experiment and the gels/blots were processed in parallel. This experiment was repeated more than three times independently with similar results. **d** Distribution of Log$_2$(fold-change) of expressed genes in RNA-seq data in Hu5/KD3 cells on each day of differentiation relative to Day 0, shown as violin and box plots. For the box plot, center lines indicate median, box limits indicate upper and lower quartiles, whiskers indicate 1.5× interquartile range and points indicate outliers; *n* = 58,278 genes. **e** Heat map of RNA-seq transcriptome analysis of differentiation markers, SELENON, hnRNP C, ADAR1, UGA/Sec recoding factors, and ER stress markers on the indicated days in differentiating Hu5/KD3 cells. Colors correspond to the per-gene z-score calculated from log$_{10}$ fragments per million (FPM) reads mapped. **f** RT-qPCR to estimate the Alu exonization level of *SELENON* mRNAs during myoblast differentiation. Transcripts of each isoform were quantified using Hu5/KD3 total RNA from each day of differentiation. The graph represents the ratio between the inclusion and skipping isoforms. Data are presented as mean values ±S.D.; *n* = 3 biologically independent samples. Source data and unprocessed gel images are provided in the Source Data file.

Northern blotting revealed that the steady-state level of tRNA$^{Sec}$ did not significantly change over the course of myoblast differentiation (Fig. 4c). Next, we carefully analyzed RNA-seq data of the recoding factors involved in Sec insertion (Supplementary Fig. 1) and identified several recoding factors that were

downregulated at the late stage (Fig. 2e). A volcano plot comparing the datasets from Day 0 and 5 (Fig. 4d) revealed that *SEPHS1*, *SEPHS2*, *PSTK TRNAU1AP*, and *EIF4A3* are down-regulated during the late stage. Although *SEPHS1* encodes selenophosphate synthetase, which is thought to recycle Sec to

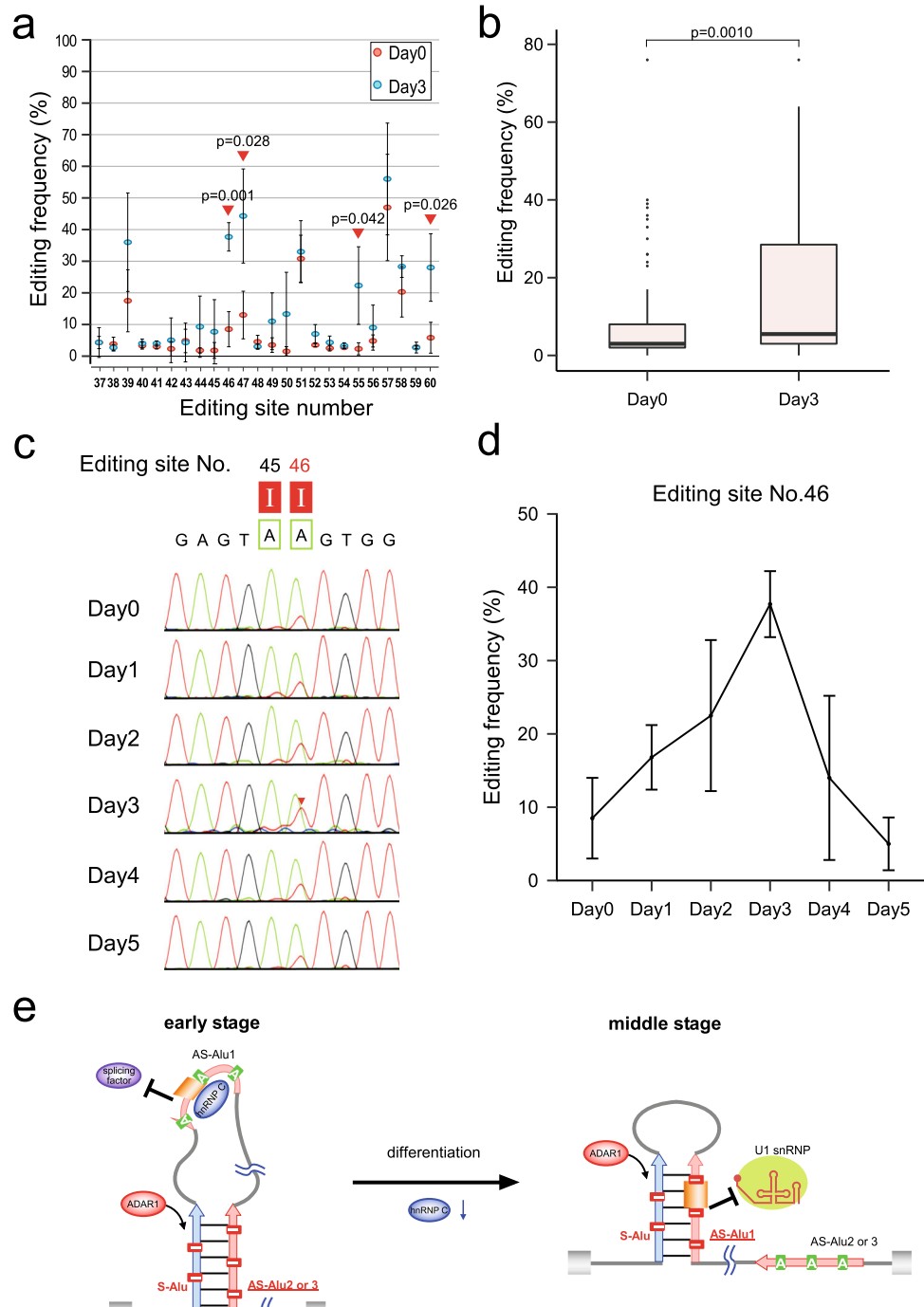

**Fig. 3 Dynamic alteration of A-to-I RNA editing during myoblast differentiation. a** Alteration of A-to-I RNA editing frequency in AS-Alu1 from Day 0 (red) to 3 (blue). Data are presented as mean values ±S.D. The positions showing statistical significance [$p < 0.05$, determined by unpaired two-tailed $t$ test; $n = 4$ (Day 0) and 3 (Day 3) biologically independent samples] are indicated by red triangles. Source data are provided in Source Supplementary Data 1. **b** The distribution of the A-to-I RNA editing frequency in AS-Alu1 on Days 0 and 3 is shown as box plots (center lines indicate median, box limits indicate upper and lower quartiles, whiskers indicate 1.5× interquartile range and points indicate outliers). Error bars denote S.D. Statistical significance was determined by unpaired two-tailed $t$ test; $n = 96$ (Day 0) and 72 (Day 3). Source data are provided in Source Supplementary Data 1. **c** Sanger sequences of the 5′SS of the Alu exon in AS-Alu1 of *SELENON* mRNAs in differentiating Hu5/KD3 cells on the indicated days. The A-to-I RNA editing sites are boxed, with the site number shown above. **d** Frequency of editing site No. 46 in Hu5/KD3 cells on different differentiation days. Data are presented as mean values ±S.D.; $n = 4$ (Day 0 and Day 2), 5 (Day 1), and 3 (Day 3–5) biologically independent samples. Source data are provided in Source Supplementary Data 1. **e** Schematic model of conformational changes in *SELENON* pre-mRNA from the early-to-middle stage of differentiation. At the early stage, hnRNP C binds to AS-Alu1, and then S-Alu forms a double-stranded structure with AS-Alu2 or 3. A-to-I RNA editing occurs in this double strand. At the middle stage, the level of hnRNP C decreases. AS-Alu1 forms a double-stranded structure with S-Alu, which is recognized by ADAR1, which then performs A-to-I RNA editing in this double-stranded region, including the 5′SS of the Alu exon, thereby inhibiting U1 snRNA recognition. Source data and unprocessed gel images are provided in Source Data file.

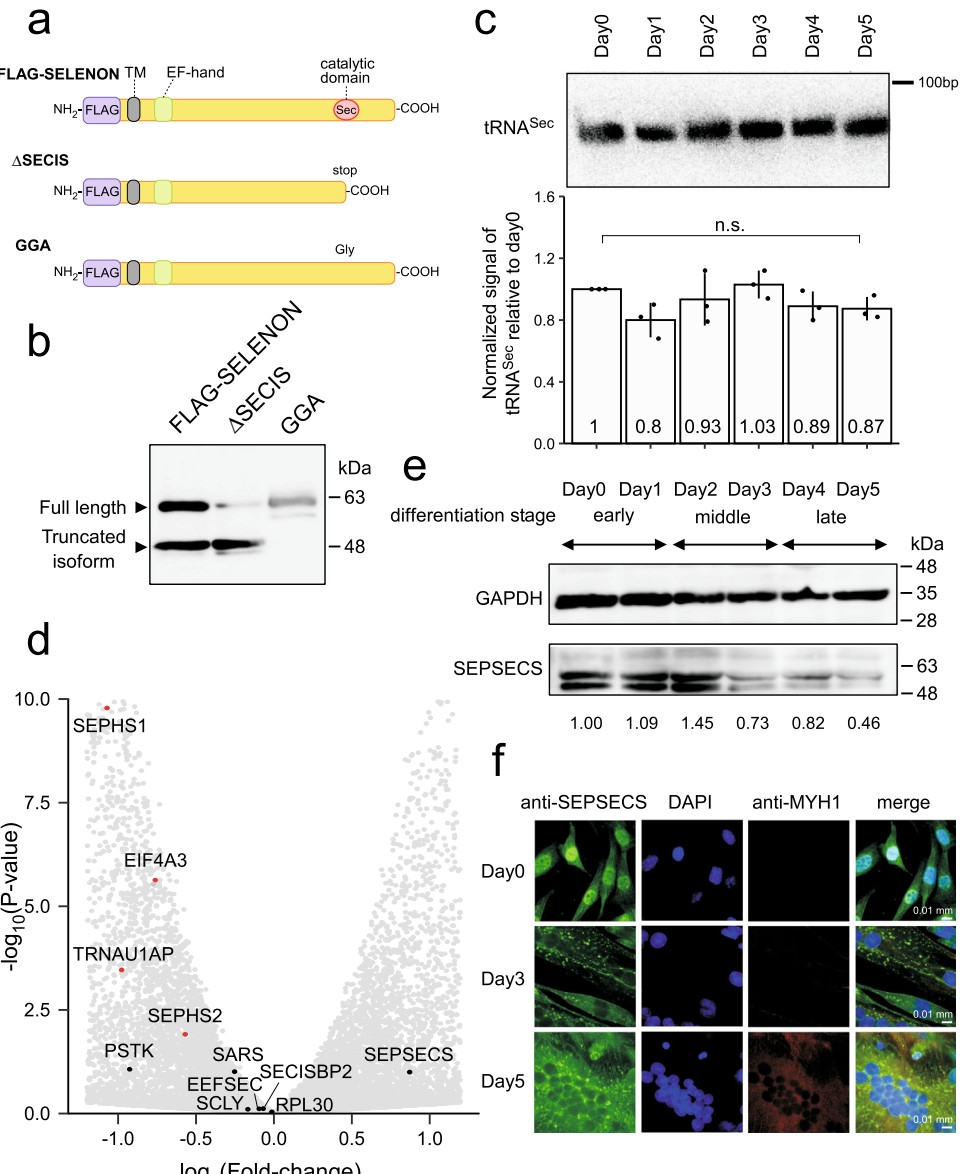

**Fig. 4 Dynamic regulation of UGA/Sec recoding during myoblast differentiation. a** Constructs of the FLAG-SELENON expression vector and its mutants ΔSECIS and GGA. TM (transmembrane domain), EF-hand domain, and Sec residue in the catalytic domain are indicated. **b** Expression of FLAG-SELENON and its mutants in HEK293T cells, detected by western blotting with an anti-FLAG antibody. This experiment was done once. **c** Northern blotting of cytoplasmic tRNA$^{Sec}$ in total RNA obtained on each day of differentiation. Relative steady-state level of tRNA$^{Sec}$ on each day of differentiation compared with the level on Day 0 is shown below. Results were normalized relative to the band intensity of 5.8 S rRNA in the same sample. Data are presented as mean values ±S.D. Statistical significance was determined by unpaired two-tailed $t$ test; $n = 3$ biologically independent samples. **d** Volcano plot of RNA-seq transcriptome data showing altered gene expression [log$_2$(fold-change)], with $p$ values, on Day 5 versus Day 0 of Hu5/KD3 cell differentiation. UGA/Sec recoding factors are indicated, and significantly differentially expressed genes (two-sided $p$ value <0.05 determined by likelihood ratio test) are highlighted in red. **e** Expression analyses of *GAPDH* (control) and *SEPSECS* in differentiating Hu5/KD3 cells by western blotting on the indicated days. The expression level of *SEPSECS* was normalized to that of *GAPDH*, making the relative expression level on Day 0 = 1.00 (shown below). The samples derived from the same experiment and the gels/blots were processed in parallel. This experiment was repeated more than three times independently with similar results. **f** Subcellular localization of SEPSECS on the indicated days in differentiating Hu5/KD3 cells immunostained with an anti-SEPSECS antibody (green). Nuclei were stained with DAPI (blue) and myotubes were stained with an anti-myosin antibody (red). Magnification: ×100. Exposure time: 150 ms (Day 0) or 300 ms (Day 3 and 5). This experiment was done once. Source data and unprocessed gel image are provided in Source Data file.

generate selenophosphate[64], the function of *SEPHS1* in seleno-protein synthesis remains unclear because targeted inactivation of *Sephs1* in mouse liver did not reduce selenoprotein expression[65]. Similarly, *Trnau1ap* inactivation did not alter selenoprotein expression in mouse liver[66]. RNA-seq data showed that two essential Sec recoding factors, *SEPHS2* and *PSTK*, decreased during differentiation, leading to downregulation of *SELENON* expression during the late stage of differentiation. Moreover, the

steady-state levels of SEPSECS protein were reduced on Day 3 to 5 (Fig. 4e), although this was not reflected by a corresponding change in the mRNA level (Figs. 2e, 4d), indicating that SEPSECS is downregulated post-transcriptionally during the late stage of myoblast differentiation. SEPSECS converts phosphoseryl (pSer)-tRNA$^{Sec}$ to Sec-tRNA$^{Sec}$ using selenophosphate as a Se donor[67] (Supplementary Fig. 1). In addition, immunostaining analysis of the subcellular localization of SEPSECS during myoblast

differentiation (Fig. 4f), showed that on Day 0, SEPSECS was predominantly localized in the nucleus (as shown by co-staining with DAPI), but was also present in the cytoplasm. On Day 3, myoblasts were differentiating into myotubes, as determined by co-staining for MYH chain (Fig. 4f). The nuclear signal of SEPSECS was markedly reduced on Day 3 (Fig. 4f) and 5 (Fig. 4f). Therefore, subcellular localization of SEPSECS differs according to myoblast differentiation. Taken together, these observations indicate that Sec-tRNA$^{Sec}$ formation is downregulated by reduced expression of recoding factors, including *SEPHS2*, *PSTK,* and *SEPSECS*, during the late stage of differentiation.

**Reduction of Sec-tRNA$^{Sec}$ formation in the late stage of myoblast differentiation**. Given that essential components in the recoding machinery were downregulated at the late stage, we investigated whether the Sec-tRNA$^{Sec}$ level was actually reduced during myoblast differentiation. To this end, we directly analyzed amino-acid species attached to tRNA$^{Sec}$ isolated from myoblasts and myotubes (Fig. 5a). We first prepared total aminoacyl-tRNAs from Hu5/KD3 cells harvested on Day 0 and 5; tRNA was isolated by phenol extraction under acidic conditions to avoid deacylation. To stabilize the amino-acid moieties attached to tRNAs, the amino groups were acetylated, and the selenol group was alkylated. Then, the aminoacylated tRNA$^{Sec}$ (Fig. 5b) was successfully isolated by reciprocal circulating chromatography (Fig. 5c)[68], followed by RNase T$_1$ digestion. The RNA fragments of the isolated tRNA$^{Sec}$ were analyzed by capillary liquid chromatography (LC)-nano-electrospray ionization (ESI)-mass spectrometry (RNA-MS) (Fig. 5d and Supplementary Data 2)[69]. Most of the RNase T$_1$-digested fragments were detected as negative ions.

Regarding the 3′ terminal fragment, we detected three species of CCA fragments attached to different amino acids: Ser (CCA-Ser+Ac), pSer (CCA-pSer+Ac), and Sec (CCA-Sec+Ac+Alk) (Fig. 5e). These fragments were further probed by collision-induced dissociation (Fig. 5f and Supplementary Fig. 16). Because Se has several major natural isotopes, including $^{80}$Se (49.61%), $^{78}$Se (23.77%), $^{76}$Se (9.37%), $^{82}$Se (8.73%), and $^{77}$Se (7.63%), we observed a unique pattern in the mass spectrum (Fig. 5g) identical to the theoretical mass spectrum generated by isotope simulation (Supplementary Fig. 17). Judging from the peak intensities of the mass chromatograms of aminoacylated CCA fragments, we estimated the fraction of each species. On Day 0, pSer-tRNA$^{Sec}$ (54.1%) was the most abundant, and Sec-tRNA$^{Sec}$ (33.5%) was the second most abundant; the remaining tRNA$^{Sec}$ (12.4%) was charged with Ser (Fig. 5e). On Day 5, the proportion of Sec-tRNA$^{Sec}$ plunged to 14.4%, whereas the proportion of pSer-tRNA$^{Sec}$ increased to 77.7% (Fig. 5e), indicating that the conversion process from pSer to Sec was impaired at the late stage of myoblast differentiation. This finding is nicely explained by our observation that SEPSECS and some UGA/Sec recoding factors were downregulated during the late stage (Fig. 4d, e, and Supplementary Fig. 1).

Moreover, tRNA$^{Sec}$ contained several post-transcriptional modifications (Fig. 5b)[70,71]. If the modifications in the anticodon region are regulated, the hypomodified tRNA$^{Sec}$ could affect its recoding activity. To test this, we carefully measured modification frequency by RNA-MS. In human tRNA$^{Sec}$, 5-methoxycarbonylmethyluridine (mcm$^5$U) and its 2′-$O$-methyl derivative (mcm$^5$Um) are present at position 34, and $N^6$-isopentyladenosine (i$^6$A) is found at position 37 in the anticodon loop (Fig. 5b). These modifications contribute to stabilization of the codon–anticodon interactions. Because the relative frequencies of mcm$^5$U34 and mcm$^5$Um34 are affected by Se availability[72,73], we analyzed modification status in the anticodon region of tRNA$^{Sec}$ between Day 0 and 5, and found that the level

of mcm$^5$Um34 was slightly higher on Day 5 than on Day 0 (Supplementary Fig. 18) probably due to high Se concentration in the medium on Day 5. This observation rules out the possibility that *SELENON* downregulation is caused by hypomodification of tRNA$^{Sec}$ during the late stage of myoblast differentiation.

The findings described in this section demonstrate a reduction of Sec-tRNA$^{Sec}$ formation in myotubes. Judging from the distance of the UGA/Sec codon from the exon–exon junctions, the UGA/Sec codon is a PTC[18]. The reduction in the level of *SELENON* mRNA at the late stage (Fig. 2e, and Supplementary Fig. 11) could be partly explained by NMD resulting from recoding regulation.

## Discussion

During skeletal myogenesis, the timely expression of lineage-specific genes is precisely and dynamically regulated by transcription, epigenetic factors, and miRNAs. During myoblast differentiation, Ca$^{2+}$ signaling plays important role in regulating transcription factors, Ca$^{2+}$-dependent kinases, and phosphatases[74,75]. For example, MYOG, a key factor involved in myoblast differentiation, is induced by activation of calcineurin upon elevation of cytoplasmic Ca$^{2+}$ concentrations[76]. Cytoplasmic Ca$^{2+}$ is regulated by the ER, an intracellular Ca$^{2+}$ reservoir. During myoblast differentiation, ER Ca$^{2+}$ is transiently released into the cytoplasm through IP$_3$R[77]. At this stage, SELENON becomes activated by sensing low-ER Ca$^{2+}$ concentration via its EF-hand domain and chemically reduces proteins on the ER membrane, including the ER Ca$^{2+}$ import pump SERCA[40]. The reduced form of SERCA facilitates Ca$^{2+}$ influx into the ER. Hence, the steady-state level of SELENON must be maintained during the early-to-middle stages of myoblast differentiation.

The ER also plays a critical role in protein folding, processing, glycosylation, and other processes. Reduction of the ER Ca$^{2+}$ concentration leads to ER stress by downregulating protein folding enzymes such as disulfide isomerases, and molecular chaperones such as BiP and calreticulin, which use Ca$^{2+}$ as a cofactor[78–80]. Prior to myoblast cell fusion, ER stress is transiently induced by ER Ca$^{2+}$ depletion and plays a critical role in myoblast differentiation[81,82]. Indeed, RNA-seq data confirmed that ER stress markers were elevated in the middle stage of Hu5/KD3 differentiation. ER stress triggers the unfolded protein response (UPR) pathway, leading to the activation of ER oxidoreductin 1 (ERO1), which in turn generates ROS[83]. Although ROS plays an important role in ER protein folding, myoblasts are vulnerable to ROS-induced oxidative stress. Differentiation-incompetent myoblasts incapable of sustaining cellular stress are removed by UPR-activated apoptosis[84,85]. SELENON has high antioxidant activity and plays a role in protecting myoblasts from oxidative stress elicited by ERO1 induction[41]. In fact, *Selenon*-knockout mice exhibit dysfunction in highly active muscles triggered by a maladaptive ER stress response[43]. For these reasons, *SELENON* expression must be maintained to protect myoblasts from oxidative stress during myotube maturation, facilitating cell fusion to form strong muscle fibers. The regulation of Alu exonization of *SELENON* mRNA revealed in this study is important for maintaining the expression level of *SELENON* from the early-to-middle stage of myoblast differentiation. Aberrant regulation of Alu exonization could impair the sophisticated regulation of *SELENON* expression, leading to muscle atrophy due to a decreased number of fusing myoblasts.

In mature skeletal muscle, ER Ca$^{2+}$ release and uptake play important roles in muscle contraction and relaxation. To facilitate this process during myoblast differentiation, the ER is converted to SR, which is capable of more efficient Ca$^{2+}$ release[86]. During muscle contraction, RyR1 is activated to release ER Ca$^{2+}$ into the

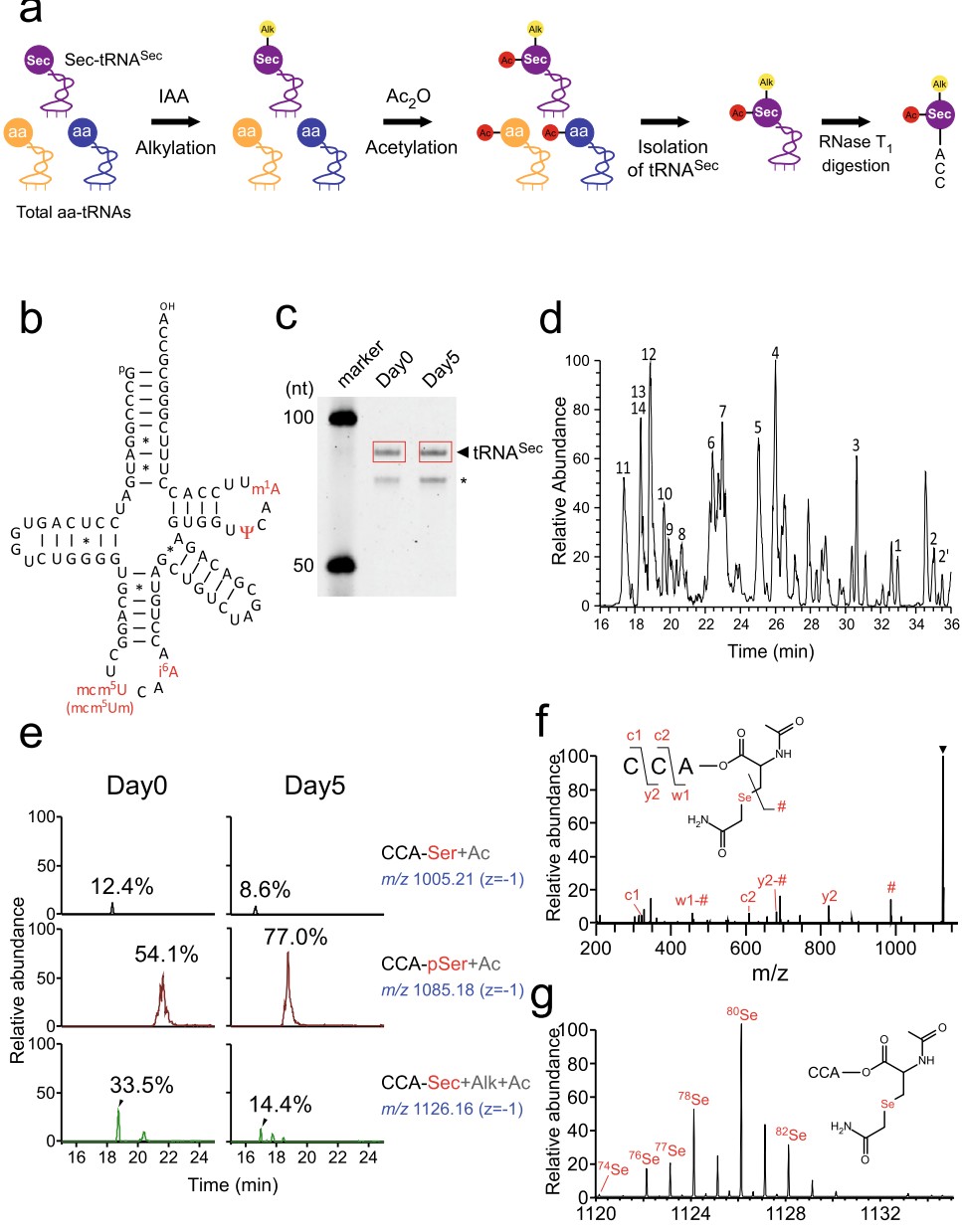

**Fig. 5 Reduction of Sec-tRNA^Sec formation at the late stage of myoblast differentiation. a** Outline of aminoacyl-tRNA^Sec preparation for LC/MS analysis. Small yellow and red circles attached to amino acids represent alkyl and acetyl groups, respectively. IAA, iodoacetamide. Ac₂O, acetic anhydride. aa, amino acid. **b** Secondary structure of human tRNA^Sec harboring post-transcriptional modifications (highlighted in red). The symbols used to denote modifications are as follows: mcm^5U, 5-methylcarboxymethyluridine; i^6A, N^6-isopentyladenosine.; Ψ, pseudouridine; m^1A, 1-methyladenosine. Watson–Crick and GU pairs are indicated by solid lines and asterisks, respectively. **c** tRNA^Sec from Day 0 and 5 of Hu5/KD3 differentiation was isolated and resolved by 10% denaturing PAGE. The excised bands for tRNA^Sec are boxed in red. The shorter band indicated by the asterisk was found to be tRNA^Arg, which was co-isolated. This experiment was done once. The unprocessed gel image is provided in Source Data file. **d** Base peak mass chromatogram for tRNA^Sec digested by RNase T₁. Assigned fragments are numbered and listed in Supplementary Data 2. **e** Mass spectrometry analyses of amino acids attached to the CCA end of tRNA^Sec isolated on Day 0 and 5 of Hu5/KD3 differentiation. Extracted ion chromatograms (XICs) show corresponding negative ions of the CCA trinucleotides, with amino acids indicated on the right-hand side of each chromatogram. The m/z value and charge state are shown on the right. Frequency was calculated from the relative peak intensities of the aminoacylated fragments. **f** Collision-induced dissociation spectrum of the CCA trinucleotides with Sec bearing alkylation (Alk) and acetylation (Ac). Product ions were assigned as indicated on the sequence. **g** Unique MS isotope distribution of Sec-attached CCA trinucleotides from the isolated tRNA^Sec at Day 0. The chemical structure of the molecule is shown on the panel. The major natural isotopes of Se are shown on each spectrum.

cytoplasm[87], with this transient increase in cytosolic Ca^2+ activating the myoplasmic Ca^2+ buffering system and the contractile apparatus[88]. This process is called excitation-contraction coupling (EC-coupling)[89]. Subsequently, the cytosolic Ca^2+ is retrieved by the SR through SERCA[90,91]. Because SELENON

activates SERCA and increases its Ca^2+ pumping ability, hyperactive SERCA with a high level of SELENON expression dysregulates Ca^2+ flux and impairs EC-coupling in skeletal muscle. The present study found that the downregulation of SELENON during the late stage of myoblast differentiation is caused primarily

by Alu exonization mediated by reduction of both hnRNP C and ADAR1, and downregulation of UGA/Sec recoding. Normal muscle contraction and relaxation might be maintained by the post-transcriptional regulation of *SELENON* expression.

This study found that the Alu exonization of *SELENON* mRNA was sequentially regulated by hnRNP C and ADAR1 during the early-to-middle stage of myoblast differentiation. The steady-state level of hnRNP C decreases from the early-to-the-middle stage, and ADAR1 decreases during the middle stage. Although the mechanism that regulates hnRNP C remains unknown, the expression of *ADAR1* is suppressed by muscle-specific miR-1 and miR-206 during myoblast differentiation[61,92]. Because the Alu exon of *SELENON* mRNA accumulates markedly in mature skeletal muscle (Supplementary Fig. 2)[46], we investigated the relationships between the Alu exonization ratio of *SELENON* mRNA and the expression levels of *ADAR1* and *hnRNP C* in individual samples from ten tissues (brain, colon, heart, kidney, lung, muscle, ovary, pancreas, stomach, and testis) using RNA-seq data from the GTEx project[51]. Six samples of each tissue from young, middle-aged, and elderly subjects, with one man and one woman in each age category, were chosen (Supplementary Data 3). The percent splice-in (PSI) values of the *SELENON* Alu exon were determined by counting the reads that map to Exon2–Alu exon, Alu exon–Exon3, and Exon2–Exon3[93]. All six skeletal muscles samples exhibited lower steady-state levels of *ADAR1* and higher Alu exonization ratios than the other tissues (Supplementary Fig. 19a). Moreover, we observed a weak negative correlation ($r = −0.46$) between the PSI value of *SELENON* Alu exon and the level of *ADAR1* in the individual samples (Supplementary Fig. 19a). In contrast, we observed little correlation between Alu exonization and the level of *hnRNP C* in individual samples (Supplementary Fig. 19b). These observations suggest that, not only in skeletal muscle but also in other tissues, Alu exonization of *SELENON* mRNA is mainly regulated by A-to-I RNA editing.

Alternative splicing is frequently modulated by A-to-I RNA editing[94,95]. ADAR2 regulates its expression by editing the intronic region of its own mRNA from AA to AI, which serves as a splice acceptor to include the cryptic exon, resulting in aberrant mRNA[96]. A-to-I RNA editing of *GPR81* mRNA generates a potential splice donor from AU to IU[97]. The Alu exon of *LUSTR1* mRNA is excluded by editing the splice acceptor from AG to IG[97]. A comprehensive search of A-to-I RNA editing sites in the consensus sequence for pre-mRNA splicing revealed that about 500 editing sites are present at splice acceptor sites[98]. This study also found A-to-I RNA editing at the +4 intronic position of 5′SS in 13 exons, including the Alu exon in *SELENON*, although none of them has been examined experimentally. To the best of our knowledge, the *SELENON* Alu exon is the first experimental demonstration of an mRNA in which A-to-I RNA editing at the U1 snRNA recognition site regulates mRNA splicing.

Intronic Alu elements have cryptic splicing signals. A part of the Alu element is occasionally exonized and incorporated into mature mRNA[47]. Activation of cryptic exons frequently causes human diseases; in 25% of such cases, the cryptic exons are derived from the antisense strand of Alu elements[99,100]. By contrast, exonization of Alu elements is thought to have played a role in generating new exons over the course of primate genome evolution[101]. ADAR1 recognizes long dsRNA structures formed by Alu elements and introduces inosines into this structure. More than 90% of the A-to-I RNA editing sites in the human transcriptome reside in Alu elements[54,55]. ADAR1-mediated editing might modulate species- and/or tissue-specific gene expression by regulating Alu exonization[54]. A comparison of the Alu exons of *SELENON* mRNA in humans, macaques, and chimpanzees at the transcriptome level[46] found that the Alu exon was absent from

the corresponding region in macaques. Although the inclusion isoform of the Alu exon was detected in chimpanzees, it was much less abundant than the human isoform. In addition, the RNA editing frequency is significantly higher in humans than in chimpanzees and macaques[102]. These observations suggest that muscle-specific Alu exonization of *SELENON* mRNA is a regulatory mechanism specific to humans. Future studies may identify pathogenic mutations that cause dysregulation of Alu exonization in *SELENON* mRNA.

We also found that the UGA/Sec recoding efficiency of *SELENON* is downregulated due to a reduction in the levels of Sec-tRNA$^{Sec}$ in the middle-to-late stage of myoblast differentiation. This is the first reported instance of gene expression regulation mediated by recoding regulation in any biological context. Similar recoding regulation may be involved in other biological events.

In the middle-to-late stage, *SELENON* mRNA can be degraded by NMD and the short isoform of SELENON accumulates. The short isoform is a truncated SELENON bearing TM and EF-hand, but lacking a catalytic domain (Supplementary Fig. 14). *SELENON* mutants in which the Sec recoding site has been replaced with Cys or Ser are unable to rescue ER Ca$^{2+}$ uptake of *SELENON*-deficient cells[103], demonstrating the necessity of the active site for SELENON function. However, the short isoform can localize to the ER, and it is quite stable during myoblast differentiation (Fig. 2c), indicating that it has some biological function.

The recoding system for Sec insertion consists of complicated mechanisms, with several recoding factors dynamically participating in Sec-tRNA$^{Sec}$ formation and its recruitment to the recoding site on the ribosome. Many of these mechanisms, however, remain undetermined, although a supramolecular complex model has been suggested[104]. Sec-tRNA$^{Sec}$ that forms in the cytoplasm associates with SEPHS1, SEPSECS, and TRNAU1AP to form a large ribonucleoprotein complex, which is imported into the nucleus, where EEFSEC replaces SEPSECS in the complex (Supplementary Fig. 1)[104–107]. However, the functional roles of SEPHS1 and TRNAU1AP in Sec insertion remain unclear[65,66]. The Sec-tRNA$^{Sec}$-EEFSEC complex interacts with the SECIS element of mRNA with the help of SECISBP2, followed by export to the cytoplasm where it is translated for Sec insertion (Supplementary Fig. 1). The present study found that the nuclear signals of SEPSECS were reduced during the late stage of differentiation (Fig. 4f), suggesting that the nuclear ribonucleoprotein of the Sec insertion machinery is less abundant during the middle and late stages.

Among the type 1 selenoproteins other than SELENON, GPX1 is also downregulated during myoblast differentiation at both the mRNA (Supplementary Fig. 15a) and protein levels (Supplementary Fig. 15b). *GPX1* mRNA may be degraded by NMD due to a reduction in Sec-tRNA$^{Sec}$ formation during the late stage of myoblast differentiation. Transient ROS production during the middle stage of myoblast differentiation activates NF-κB, which suppresses *MYOD1* expression[9,108,109]. Thus, GPX1 may play a critical role in myoblast differentiation by reducing ROS during the middle stage to maintain the MYOD1 level[110–112]. In the late-stage and mature muscles, both GPX1 and MYOD1 levels decrease, but these two proteins can be activated immediately in response to exercise-induced oxidative stress[113,114].

This study revealed the mechanisms involved in post-transcriptional regulation of *SELENON* expression during myoblast differentiation (Fig. 6). At the early stage, hnRNP C binds to AS-Alu1 of *SELENON* mRNA and inhibits recognition by U2AF65, thereby preventing Alu exonization. In the middle stage, as hnRNP C decreases, and AS-Alu1 is released from hnRNP C to form a long double strand with S-Alu, serving as a substrate for ADAR1. ADAR1 is responsible for A-to-I RNA editing around

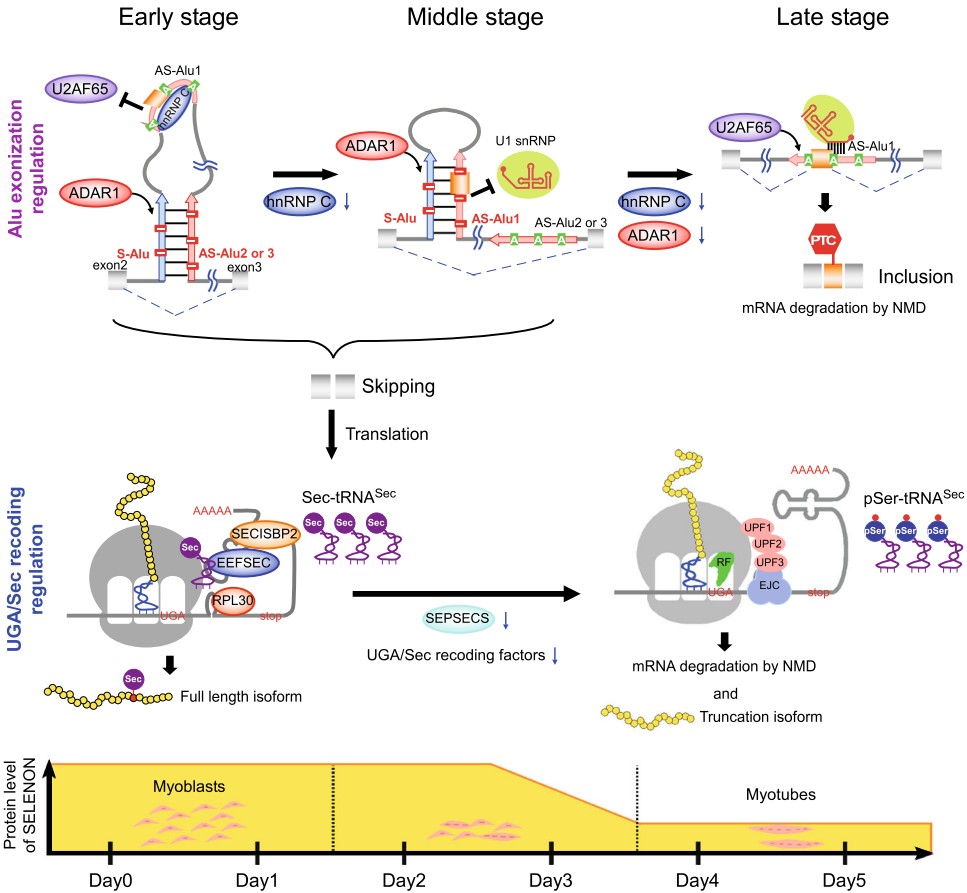

**Fig. 6 Mechanisms underlying post-transcriptional regulation of *SELENON* expression during myoblast differentiation.** At the early stage, hnRNP C binds to AS-Alu1 of *SELENON* pre-mRNA and inhibits U2AF65 recognition, thereby suppressing Alu exonization. At the middle stage, the level of hnRNP C decreases, thereby releasing AS-Alu1, which forms a long double strand with S-Alu to be recognized by ADAR1. ADAR1 edits AS-Alu1, including the 5′SS of the Alu exon, thereby inhibiting U1 snRNA recognition to suppress Alu exonization. Sequential regulation of Alu exonization by the two consecutive mechanisms maintains *SELENON* expression from the early-to-middle stage of differentiation. At the late stage, the level of ADAR1 also decreases, thereby promoting Alu exonization and generating aberrant mRNA with PTC, followed by degradation by NMD. Moreover, at the middle-to-late stage, the levels of SEPSECS and other UGA/Sec recoding factors decrease to downregulate Sec-tRNA^Sec formation, thereby suppressing UGA/Sec recoding. As a result, *SELENON* mRNA is subjected to degradation (probably by NMD) or translation to yield the truncated isoform of SELENON.

the 5′SS of the Alu exon, preventing U1 snRNA recognition and suppressing Alu exonization. Consecutive regulation of Alu exonization by these two distinct mechanisms maintains *SELE-NON* expression from the early-to-middle stage of myoblast differentiation. In the middle-to-late stage, the level of ADAR1 also decreases and Alu exonization is more frequent, resulting in the production of an aberrant mRNA that is degraded by NMD. At this stage, the level of Sec-charged tRNA^Sec is reduced due to downregulation of SEPSECS and other UGA/Sec recoding factors, including SEPHS2 and PSTK. As a result, *SELENON* mRNA is degraded by NMD, generating the truncated isoform of SELE-NON. Further studies will be necessary to elucidate the physiological importance of these findings. The resultant knowledge could contribute to effective treatment of SELENON-RM or prevention of muscle loss with aging.

## Methods

**Source of human tissue RNA.** Total RNA from adult human brain (A507293), colon (A508349), heart (A809126), kidney (A801018), liver (A507018), lung (A709150), ovary (A709150), pancreas (A905068), stomach (A507057), testis (A504299), and skeletal muscle (B207204) were purchased from BioChain Institute (Newark, CA, USA).

**Cell culture.** HeLa (CCL-2, ATCC) and HEK293T (CRL-11268, ATCC) cells were cultured in high-glucose Dulbecco's modified Eagle's medium (D5796, Sigma) with

10% fetal bovine serum (Gibco) and 1% penicillin–streptomycin (Fujifilm Wako Pure Chemical Corporation) in a 5% $CO_2$ humidified incubator at 37 °C.

Immortalized human myogenic cells (Hu5/KD3)[57,58] were kindly provided by Dr. Naohiro Hashimoto (National Center for Geriatrics and Gerontology, Japan). Hu5/KD3 cells were cultured in primary myocyte growth medium (pmGM), which consists of high-glucose DMEM containing 20% FBS, 2% Ultroser G serum substitute (PALL), and 1% penicillin–streptomycin, in a 5% $CO_2$ humidified incubator at 37 °C. Culture dishes were manually coated with collagen using Cellmatrix Type I-P (Nitta Gelatin).

**Myoblast differentiation and RNA extraction.** Two days before differentiation, Hu5/KD3 cells were seeded in dishes ($3.0 \times 10^5$ cells per 60 mm dish, or $8.0 \times 10^5$ cells per 100 mm dish). On Day 0, the medium was replaced with primary myocyte differentiation medium (pmDM), which consists of high-glucose DMEM supplemented with 2% FBS, 5 μg/mL bovine holo-transferrin bovine (Sigma), 10 μg/mL insulin (Roche), and 10 nM sodium selenite (Sigma). On Day 2, cytarabine (Tokyo Chemical Industry) was added (10 μM) to remove proliferating cells. On Day 4, the medium was replaced with pmDM to remove cytarabine. Cell images of differentiating myoblasts were obtained by IncuCyte S3 Live-Cell Analysis System (Sartorius). Total RNA was extracted from cells harvested on each day of differentiation using TriPure Isolation Reagent (Roche).

**Western blotting and antibodies.** Differentiating Hu5/KD3 cells in 60 mm dish were lysed in 100 μL RIPA + EDTA buffer (50 mM Tris-HCl [pH 8.0], 150 mM NaCl, 1% NP-40, 0.5% sodium deoxycholate, 0.1% SDS, 1 mM EDTA, 1 mM DTT, and 1× Protease Inhibitor [cOmplete, EDTA-free; Roche]). The lysates were run on 10% SDS-PAGE gels and electroblotted onto a PVDF membrane (Amersham Hybond P; GE Healthcare) using a Transblot Turbo apparatus (Bio-Rad). Western blotting was performed with the following primary antibodies: anti-ADAR1 (1:500)

(sc-271854, Santa Cruz Biotechnology), anti-ADAR2 (1:500) (sc-73409, Santa Cruz Biotechnology), anti-DYKDDDDK (FLAG) (1:6000) (014-22383, Wako), anti-GAPDH (1:1000) (AM4300, Invitrogen), anti-GPX1 (1:500) (sc-133160, Santa Cruz Biotechnology), anti-hnRNP C1/C2 (1:1000) (sc-32308, Santa Cruz Biotechnology), anti-MYH1 (1:100) (MF20-S, DSHB), anti-MYOG (1:2000) (sc-576, Santa Cruz Biotechnology), anti-SELENON (1:1000) (sc-365824, Santa Cruz Biotechnology), anti-SEPSECS (1:1000) (A7103, ABclonal), and anti-RENT1 (UPF1) (1:1000) (A300-038A, BETHYL). Peroxidase affinipure donkey anti-mouse/rabbit IgG (1:20000) (715-035-150/715-035-152, Jackson ImmunoResearch) and peroxidase affinipure bovine anti-goat IgG (1:20000) (805-035-180, Jackson ImmunoResearch) were used as the secondary antibodies. Proteins were detected using Pierce ECL Plus Western Blotting Substrate (Thermo Fisher Sciences) and visualized with ImageQuant LAS4000 mini (GE Healthcare). Bands were analyzed using ImageJ software (National Institutes of Health).

**RNA interference.** HeLa cells ($5 \times 10^5$ cells in 60-mm dishes) were reverse-transfected with siRNAs targeting *ADAR1* (1 nM), *ADAR2* (1 nM), *hnRNP C* (10 nM), *UPF1* (1 nM), and *luciferase* (5 nM, for control) using 5 (for single knockdown) or 10 μL (for double knockdown) of Lipofectamine RNAi MAX (Invitrogen). The sequences of the siRNAs are listed in Supplementary Data 4. Mock transfection was performed by the same procedure in the absence of siRNA. Twenty-four hours after reverse transfection, the siRNAs were forward-transfected. Forty-eight hours after the forward transfection, total RNA was extracted using the TriPure Isolation Reagent (Roche).

Knockdown efficiency was calculated by quantitative RT-PCR (RT-qPCR). Total RNAs (1 μg) were treated with 1 unit of RQ1 DNase (Promega) at 37 °C for 30 min in a 10 μL reaction mixture, followed by the addition of RQ1 DNase stop solution. Reverse transcription was performed according to the manufacturer's instructions (EvoScript Universal cDNA Master; Roche) in a 20 μL reaction mixture. Each 10 μL reaction mixture for qPCR contained 0.5 μL of template cDNA solution and KAPA SYBR FAST qPCR Master Mix (2×) Kit (Kapa Biosystems), with amplifications performed on a LightCycler 480 Real-Time PCR System (Roche). The thermal cycling conditions consisted of 40 cycles of denaturation at 95 °C for 10 s, annealing at 60 °C for 20 s, and extension at 72 °C for 5 s. The sequences of the qPCR primers are shown in Supplementary Data 4. For Fig. 1d, knockdown efficiencies validated by RT-qPCR were 90.6% (*ADAR1* KD) 92.8% (*ADAR2* KD), 92.2% (*hnRNP C* KD) and 92.4% (*UPF1* KD). For Fig. 1e, knockdown efficiencies validated by RT-qPCR were 92% (*ADAR1* KD), 93% (*hnRNP C* KD), 86% (*UPF1* KD), 88%/84% (*ADAR1/UPF1* KD), and 91%/83% (*hnRNP C/UPF1* KD).

**Vector construction and transfection.** For the minigene reporter bearing the Alu exon of *SELENON* (Fig. 1g), the genomic region of human *SELENON* spanning exons 2 to 3 was PCR-amplified with a set of primers (Supplementary Data 4), and inserted into the *Not*I and *Xho*I sites of the pcDNA3.1/Hygro+ plasmid to yield the WT construct. The A-to-G mutation at the no. 46 site in the WT construct was introduced by QuikChange mutagenesis (Agilent), using the primers shown in Supplementary Data 4.

For the *SELENON* expression vectors (Fig. 4a), the cDNA for *SELENON* mRNA with SECIS in the 3′UTR was reverse-transcribed using specific primers (Supplementary Data 4) and SuperScript III (Invitrogen), PCR-amplified, and inserted into the pcDNA3.1/Hygro+ plasmid with an N-terminal FLAG tag by SLiCE reaction[115] to give pcDNA3.1-FLAG-SELENON. The ΔSECIS construct and GGA construct were generated from this WT construct by QuikChange mutagenesis (Agilent) using the primers shown in Supplementary Data 4.

For transfection of the minigene reporter, HeLa cells cultured in six-well plates ($1.25 \times 10^5$ cells) were transfected with 2 μg of the vector using FuGENE (3 μL) (Roche). Twenty-four hours after transfection, the cells were treated with TriPure Isolation Reagent (Roche) to extract total RNA. For transfection of the *SELENON* expression vectors, HEK293T cells were used due to better protein production. HEK293T cells cultured in 24-well plates ($1.0 \times 10^5$ cells) were transfected with 500 ng vector using polyethylenimine (4.0 μL). Forty-eight hours after transfection, cells were lysed in 100 μL RIPA + EDTA buffer.

**RT-PCR.** DNase-treated total RNAs were purified by PureLink RNA Mini Kit (Invitrogen), and 1 μg aliquots were reverse-transcribed using the SuperScript III First-Strand Synthesis System for RT-PCR (Invitrogen) in a 20.0 μL reaction mixture using a gene-specific primer (SELENON-RT-Rv) or a reporter-specific universal primer (BGH-Rv) for the minigene reporters (Fig. 1g). The resulting cDNAs were diluted with 30 μL of water, and 1 μL of each was PCR-amplified with Platinum Taq DNA polymerase (Invitrogen) in a 12.5 μL reaction mixture. Amplification conditions consisted of initial denaturation at 94 °C for 2 min, followed by, 30 cycles of denaturation at 94 °C for 30 s, annealing at 55 °C for 30 s, and extension at 72 °C for 45 s. The PCR products were run on 10% polyacrylamide gel electrophoresis (PAGE) and stained with ethidium bromide to estimate the efficiency of Alu exonization. The sequences of the primers are listed in Supplementary Data 4.

The efficiency of Alu exonization in the knockdown experiment was measured by RT-qPCR. The cDNA was synthesized by reverse transcription with the SuperScript III First-Strand Synthesis System for RT-PCR (Invitrogen) using a gene-specific primer (SELENON-RT-Rv) (Supplementary Data 4). qPCR was performed as described above. For absolute quantification, PCR products of the inclusion and skipping isoforms were purified by excising the gel bands stained with SYBR Gold (Invitrogen) and quantifying them precisely to make standard curves. The standard curves were generated in the range of 0.05–20 fM for the inclusion isoform and 0.2–200 fM for the skipping isoform. The sequences of the primers are listed in Supplementary Data 4.

The efficiency of Alu exonization during Hu5/KD3 differentiation was measured by RT-qPCR. The cDNA was synthesized by reverse transcription with EvoScript Universal cDNA Master (Roche), followed by qPCR as described above. For normalization of the data, *FKBP1A* and *BAG6* were used as references because the RNA-seq data indicated that they were not altered during myoblast differentiation. The sequences of the primers are listed in Supplementary Data 4.

The frequency of A-to-I editing at each site was estimated by the peak height ratio of A and G in the Sanger sequencing chromatogram of cDNAs[48].

**ICE method.** The ICE method was basically performed as described[48,56]. DNase-treated total RNA (12.5 μg) was dissolved in 47.5 μL CE solution [50% ethanol, 1.1 M triethylammonium acetate (pH 8.6)], mixed with acrylonitrile (final concentration, 1.6 M), and incubated at 70 °C for 30 min. As a reference, the same reaction was performed in the absence of acrylonitrile. The treated RNA was purified using the PureLink RNA Mini Kit (Ambion), followed by ethanol precipitation. Treated or non-treated RNAs (200 ng) were reverse-transcribed and amplified using the SuperScript III One-Step RT-PCR system and Platinum Taq DNA polymerase (Invitrogen) in a 12.5 μL reaction mixture. RT-PCR conditions were as follows: 55 °C for 30 min, 94 °C for 2 min, and 35 touch-down cycles of PCR (two cycles of 94 °C for 30 s, 68 °C for 30 s, 72 °C for 30 s; two cycles of 94 °C for 30 s, 66 °C for 30 s, 72 °C for 30 s; three cycles of 94 °C for 30 s, 64 °C for 30 s, 72 °C for 30 s; three cycles of 94 °C for 30 s, 62 °C for 30 s, 72 °C for 30 s; 25 cycles of 94 °C for 30 s, 60 °C for 30 s, 72 °C for 30 s). The PCR products were diluted 80-fold with water and then amplified again with KOD Plus Neo (TOYOBO). The second PCR conditions were as follows: 94 °C for 2 min, followed by 25 cycles of PCR (94 °C, 30 s; 60 °C, 30 s; and 72 °C, 45 s). Primers were designed to amplify a region 300–500 bp in length, including the target inosine sites. The primer sets for the nested second PCR were designed inside the region amplified by the first round of PCR. The second PCR primers were also used for sequencing. The sequencing data were analyzed by CodonCode aligner 2.0.6 and the detected editing sites were mapped on the human genome by IGV 2.8.6. The sequences of the PCR primers are listed in Supplementary Data 4.

**RNA-seq.** Poly(A)+ RNA was prepared as described[56]. A cDNA library of poly(A)+ RNAs was prepared using the NEBNext Ultra II Directional RNA Library Prep Kit for Illumina. Seven cycles of PCR amplification were used for index addition and library fragment enrichment. Sequencing was performed for a single-read (150 bp) using the HiSeq4000 (Illumina) at the Vincent J. Coates Genomics Sequencing Laboratory (UC Berkeley). The reads were processed by Trimmomatic version 0.35[116], and aligned using STAR 2.5.3a[117] against hg38 human genome as a reference. The counts were determined by htseq-counting 0.9.1[118]. Differential expression was analyzed using edgeR 3.32.1[119].

**Northern blotting.** Total RNAs (2 μg) from differentiating Hu5/KD3 cells were dissolved by 10% denaturing PAGE, stained with SYBR Gold (Invitrogen), and electroblotted onto a nylon membrane (Amersham Hybond N+; GE Healthcare) in 0.5×TBE using a Transblot Turbo apparatus (Bio-Rad). DNA probes complementary to tRNA$^{Sec}$ (Supplementary Data 4) were phosphorylated with [γ-$^{32}$P] ATP (PerkinElmer Life Sciences) using T4 polynucleotide kinase (Toyobo). The membrane was UV cross-linked and hybridized with 4 pmol of 5′-$^{32}$P-radiolabeled DNA probe overnight at 55 °C in PerfectHyb solution (Toyobo). The membrane was washed three times with 1× SSC, dried, and exposed to an imaging plate (BAS-MS2040; Fujifilm). Radioactivity was visualized using an FLA-7000 imaging analyzer (Fujifilm).

**Mass spectrometry analysis of tRNA$^{Sec}$.** Total RNAs were prepared from Hu5/KD3 cells harvested on Day 0 and 5 by phenol extraction under acidic conditions to avoid diacylation of aminoacyl-tRNAs[120]. To stabilize the aminoacylated tRNA$^{Sec}$, total RNA was incubated with 10 mM DTT to reduce the selenol group of Sec at 55 °C for 30 min, followed by the addition of 300 mM NaOAc (pH 4.0) and 250 mM iodoacetamide, followed by incubation on ice for 1 h in the dark to alkylate the selenol group. The samples were precipitated with ethanol and dissolved in 300 mM NaOAc (pH 6.0), followed by the addition of 2% acetic anhydride four times, at 10 min intervals, on ice and incubation for 2 h to acetylate the amino group of the aminoacyl moiety attached to tRNAs. Total RNA was recovered again by ethanol precipitation and dissolved in 50 mM NaOAc (pH 5.0). tRNA$^{Sec}$ was isolated by reciprocal circulating chromatography as described[68] using the 5′ terminal ethylcarbamate amino-modified DNA probes (Sigma) (Supplementary Data 4).

Isolated tRNA$^{Sec}$ (10 ng) was digested with 0.005 U of RNase T$_1$ in 10 μL of 20 mM NH$_4$OAc (pH 5.3) at 37 °C for 1 h. The digests were mixed with 1/10th vol.

of 0.1 M triethylamine acetate (pH 7.0) and subjected to capillary LC/nano-electron spray ionization mass spectrometry on a linear ion trap-Orbitrap hybrid mass spectrometer (LTQ Orbitrap XL; Thermo Fisher Scientific) equipped with a splitless nanoflow high-performance LC (nano-HPLC) system (DiNa, KYA Technologies) with a nano-LC trap column (C18, $0.1 \times 0.5$ mm; KYA Technologies) and a capillary column (HiQ Sil C18W-3, $0.1 \times 100$ mm; KYA Technologies), as described[69,121]. The RNA fragments were separated for 40 min at a flow rate of 300 nL/min by capillary LC using a linear gradient from 2 to 100% solvent B (v/v) in a solvent system consisting of 0.4 M 1,1,1,3,3,3-hexafluoro-2-propanol (HFIP) (pH 7.0) (solvent A) and 0.4 M HFIP (pH 7.0) in 50% methanol (solvent B). The eluent was ionized by ESI source in a negative polarity and scanned over an $m/z$ range of 600–2000. Xcalibur 2.0.7 (Thermo Fisher Scientific) was used to operate the system. LC/MS data were analyzed using the Xcalibur Qual browser (Thermo Fisher Scientific).

**Immunostaining**. Cells were fixed with 3.7% formaldehyde, permeabilized with 1% Triton X-100, and blocked with 20% EzBlock Chemi (ATTO). Immunostaining was performed with the following primary antibodies: anti-SEPSECS (1:200) (A7103, ABclonal) and anti-MYH1 (1:200) (MF20-S, DSHB). The secondary antibodies included anti-mouse/rabbit IgG Alexa fluor 488 (1:1000) (A11029/A11008, Invitrogen) and anti-mouse IgG Alexa fluor 594 (1:1000) (A11005, Invitrogen). To visualize DNA, the cells were stained with DAPI (1:1000). Images were acquired using a DMI 6000 B (Leica).

**GTEx data processing**. The expression data of *ADAR1* and *hnRNP C* in multiple tissues were obtained from dbGaP GTEx Analysis V8 release (dbGaP Accession phs00424.v8.p2). The PSI values were calculated as described[93].

**Reporting summary**. Further information on research design is available in the Nature Research Reporting Summary linked to this article.

## Data availability
The data supporting the findings of this study are available from the corresponding authors upon reasonable request. Sequencing data were deposited in the NCBI Sequence Read Archive under BioProject ID PRJNA705848. Source data for the figures and supplementary figures are provided as a Source Data file.

## Code availability
Scripts used to analyze the data and plot the figures are available upon request.

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

## Acknowledgements

The authors thank the members of the Suzuki laboratory, particularly A. Nagao, Y. Sakaguchi, and K. Miyauchi, for experimental support and insightful suggestions and discussions; Naohiro Hashimoto (National Center for Geriatrics and Gerontology, Japan) for providing Hu5/KD3 cells; and Jernej Ule (The Francis Crick Institute) and Kathi Zarnack (Buchmann Institute for Molecule Life Sciences) for the list of Alu exons regulated by hnRNP C. We also thank Nobuyoshi Akimitsu, Yusuke Hirabayashi (UTokyo), Yoshiho Ikeuchi (UTokyo), and Masayuki Sakurai (Tokyo Univ. of Sci.) for their critical reviews and valuable comments on the manuscript. Some of this work was performed at the Vincent J. Coates Genomics Sequencing Laboratory at UC Berkeley (NIH S10 OD018174). Computations were supported by the NIG supercomputer at the ROIS National Institute of Genetics. The Genotype-Tissue Expression (GTEx) Project was supported by the Common Fund of the Office of the Director of the National Institutes of Health and by the NCI, NHGRI, NHLBI, NIDA, NIMH, and NINDS. The data used for the analyses described in this manuscript were obtained from the GTEx Portal and from dbGaP (accession number phs00424.v8.p2), both on April 1, 2020. This work was supported by Grants-in-Aid for Scientific Research (18H05272) from JSPS, and Exploratory Research for Advanced Technology (ERATO; JPMJER2002) from JST.

## Author contributions
Y.N. performed all experiments in this study, assisted by S.O. All authors discussed the results. Y.N. and T.S. wrote the paper. T.S. designed and supervised all the work.

## Competing interests
The authors declare no competing interests.
