## [Peer Review File · Nature Communications]

Title: Regulation of A-to-I RNA editing and stop codon recoding to control selenoprotein expression during skeletal myogenesisREVIEWER COMMENTS

Reviewer #1 (Remarks to the Author):

Comments:

The manuscript by Noda et al. identified post-transcriptional regulatory mechanisms of SEP11 expression during myoblast-to-myotube differentiation. They demonstrated different regulatory mechanisms from early to late stages during myoblast differentiation. Overall, the experiments were well organized.

1. The authors demonstrated that A-to-I RNA editing is dynamically altered in AS-Alu1 of SEP11 mRNA during myoblast differentiation as the steady-state level of ADAR1 gradual change, but there is no direct evidence to prove that those sites, especially mentioned site No. 46, are edited by ADAR1. It is necessary to analyze A-to-I RNA editing in AS-Alu1 of SEP11 mRNA in Hu5/KD3 cells upon overexpression and knockdown of ADAR1.
2. Sanger sequencing was used to evaluate the editing frequency at each site. The evaluation based on the peak areas by sequencing may not be accurate. I recommend the authors to carry out colony sequencing or next-generation sequencing to examine the editing frequency at each site.
3. In the middle stage, hnRNP C decreases. What's the reason?
4. In the third paragraph of "Recoding regulation suppresses SEP11 expression in the late stage of myoblast Differentiation", some Figures are not correctly marked. For example, Figure 3f, 3g, 3h should be changed to Figure 4f, 4g, 4h.
5. In Figure 5e and S17, why is the retention time of the same substance different?
6. In Figure 5e, the peaks of CCA-Sec-Alk+AC are not really like peaks. I recommend the authors to redo the experiments.

Reviewer #2 (Remarks to the Author):

Nota et al. show that an Alu-element in intron 2 of the human SELENON gene is exonized to some extent leading to NMD of the aberrant mRNA that includes a premature termination codon. The authors show that this exonization is suppressed by ADAR1-dependent A-to-I editing of the Alu-sequence and suggest that this occurs due to impaired recognition of the mRNA by U1 snRNA. Using a minigene they convincingly support this idea. HNRNP C competes with U1 snRNA and likewise suppresses exonization of the Alu sequence.

Then the authors use a human myoblast differentiation cell model and show that while SELENON pre mRNA levels are constant, SELENON mRNA levels decline in late stages of differentiation - suggesting that post-transcriptional processes account for this finding. In addition, the authors find that premature termination apparently occurs at the Sec codon and sensitivity to UPF1 knockdown suggests NMD as mechanism. The authors then go on and suggest that ADAR1 and HNRNP-C cooperate in early and mid stages of differentiation in Alu exon skipping, but their loss of expression in late myoblast differentiation accounts for the observed downregulation of SELENON mRNA and protein.

Up to there the story is flawless in the eyes of this reviewer, even if the authors address the elephant in the room, i.e. why exonization is minimal in all other cell types and organs studied, only in the discussion and supplement S15. There we learn that muscle is the organ with lowest ADAR1 and HNRNP C expression.

Then the authors find by RNASeq that many factors previously associated or co-IP'ed with selenoprotein biosynthesis are downregulated in late myoblast differentiation. While in principle the reasoning is convincing, this reviewer challenges the view that all of these observed expression changes really influence SELENON translation.

E.g. it is not so clear that SEPHS1 is "essential" for selenoprotein translation. While Tamura et al. suggested a function for Sps1 in their paper, doubts remained in the field. For example, targeted inactivation of Sephs1 in mouse liver, a well-characterized model system of selenoprotein expression, did not reduce selenoprotein expression in general, but reduced Gpx1 moderately and Selenow strongly. However, non-selenoproteins like Glrx1 and Gsto1 were massively reduced as well - questioning the specificity and the "essentiality" for selenoproteins (Tobe et al., 2016). If the high fold-change of SEPHS1 regulation in this manuscript is to be a convincing argument for its relevance for the mechanism in mind, then the high fold-change of Glrx1 in the Tobe paper should be regarded equally telling - and then there the link to selenoproteins is not bigger than to Glrx1 and Gsto1!

Similarly, in mouse liver, but also in neurons, Trnau1ap inactivation (done using two different genomic constructs) did not alter selenoprotein expression whatsoever, a finding hard to reconcile with an "essential" function (Mahdi et al., 2014).

Finally, nucleolin (NCL) was initially identified as a SECIS-binding protein, but later dismissed, at least in the sense that it does not support Sec incorporation in the in vitro translation assay as does SECISBP2. Given that the observed effect of SELENON downregulation can likely be produced by downregulation of the two undoubtedly essential factors PSTK and SEPHS2 alone, it seems premature based on the data presented to promote SEPHS1 and TRNAU1AP, possibly also NCL, as "essential" selenoprotein biosynthesis factors.

This reviewer, possibly with the majority in the field, rather considers them still in search for their real molecular function - not excluding physical interactions with selenoproteins.

SEPHS1 is called a Sec recycling factor in the manuscript, a function that seems to be occupied by Sec-lyase. At least, genetic inactivation of the Sec-lyase has the expected effect on selenoprotein expression in selenium deficiency (Byrns et al., 2014).

Similarly, after fifteen years of published research, associations have been repeatedly found, but experimental support for the function of the different "super-complexes" that Small-Howard et al reported in 2006 are not known to this reviewer. If any of these complexes relied in their function on any of the above mentioned proteins, this should have become evident by now. Thus, to interpret "a slight reduction in the signals" of factors not essential to the process and maybe involved in "supercomplexes" whose functions have not been ascertained, seems a bit stretched at the moment (Fig. 4C is well done, but a lot of speculation). This reviewer strongly suggests to stay with the solid data on editing and tRNA. If the authors want to endorse SEPHS1, TRNAU1AP, and nuclear "supercomplexes" whose functions remain elusive, they would need to show more convincing data on this. For the sake of this study on editing, these observations may not be necessary to show at all. If no compelling other data is available, the manuscript would profit massively, by simply cutting out SEPHS1 and TRNAU1AP. The data on SEPSECS is fine, convincing, and makes the point anyway.

The reduction of Sec-tRNA(Sec) in late differentiation is beautifully explained by reduction of SEPSECS (because the precursor, pSer-tRNA(Sec) is increased). Why SEPHS1 should have a hand in this - and, strangely, why SEPHS2 is not mentioned, remains unclear to this reviewer. In fact, if SEPHS1 would be limiting, how would the authors explain the increase of its downstream product pSer-tRNA(Sec)?

Discussion:

The authors may reconsider whether the details of T-tubules/Cav1.1 etc are needed or are simply elementary muscle physiology.

What might be more relevant to the main message of this paper is the answer to the thought mentioned in the discussion: "Some pathogenic mutations in SEPN1 may cause dysregulation of Alu exonization." Which?

The manuscript is well written, data are explained very clearly, experiments support the conclusions with the exceptions mentioned above. Clearly, this laboratory capitalizes on its technical championship with small RNA. Congratulations!

Why not tell the selling point earlier rather than late in the discussion: "The SEPN1 Alu exon is the first example of experimental proof of an mRNA in which A-to-I RNA editing at the U1 snRNA recognition site regulates mRNA splicing."?

The reviewer strongly suggests to stringently adhere to the unified nomenclature on selenoproteins (Gladyshev et al., 2016). So SBP2 should read SECISBP2, SepSecS SEPSECS, SEPN1 should be SELENON, Gpx1 GPX1, if referring to the protein, Gpx1, if referring to mouse mRNA etc.

Reviewer #3 (Remarks to the Author):

Review of Noda et al.,

In this manuscript the authors study the regulation of the selenoprotein SEPIN1 during myotube differentiation. The SEPIN1 pre-mRNA is modified by alternative splicing which itself is regulated by ADAR1 mediated editing and hnRNP C binding.

Evidence is provided that indicate that A to I editing at the U1 binding site will weaken the U1 interaction at the cryptic splice site.

Moreover, a short isoform of SEPIN1 occurs at the late stages of differentiation which was shown to be dependent on premature termination at the Selenocysteine codon. This process is controlled by downregulation of the selenocysteine tRNA which in turn is controlled by the factors leading to tRNA-Sec formation.

The manuscript is well written and the claims are supported by the performed experiments. However, prior to publication several points should be clarified and addressed:

- 1) Abstract: The abstract does not mention the second part of the story, the premature termination as a result of lacking tRNA Sec. Also, the abstract mentions "In the middle stage" without mentioning "of myoblast differentiation". Without the full statement, the sentence is misleading.
- 2) Page 6 first paragraph: "deficient" is spelled wrongly.
- 3) Results page 8: the experiments are started with HeLa and then switch to Hek, most likely for the better protein production in Hek cells. This should be explained.
- 4) Figure 2 c: addition of a molecular weight marker to the blot would be beneficial
- 5) Page 12 top paragraph. A reference to the tight binding of hnRNP C to the Alu element intron should be given.
- 6) Figure 4a) the schematic drawing of full length, shortened protein and suppressor codon protein is not correct (should be full length again).
- 7) Figure 4e, this blot needs quantification as the described differences are not obvious to the reader
- 8) Figure 5c: why is the 3 day timepoint missing? What is the shorter band?

Best regards,

Michael Jantsch

Reviewer #4 (Remarks to the Author):

Noda et al investigated the regulation of SEPIN1 during myogenic differentiation of the immortalized human muscle cell line Hu5/KD3. They describe a complex mechanism involving the sequential action/cooperation of alternative splicing, editing, translation and NMD. Intriguingly, data are presented

that those similar mechanisms might regulate other selenoproteins.

While the data are novel and interesting, there are a number of issues that need to be addressed as listed below.

MAJOR

1. The authors did not provide any justification for why they decided to look first at the role of Alu exonization in SEPN1 regulation. Why did they not consider simpler/more obvious mechanisms like RNA transcription or protein translation? See also point 8.

2. For their work, the authors used cell lines like HeLa and Hu5/KD3.

To support the validity of their claims, the authors must show that the regulation of SEPN1 during myogenic differentiation and the mechanisms they propose are conserved in primary muscle cells from multiple human donors.

3. Given that the authors claim that the control of SEPN1 expression they elucidated is important for the regulation of myoblast fusion and muscle contraction, it must be evolutionarily conserved. Is mouse SEPN1 regulated during *in vitro* myogenic differentiation, and *in vivo* during muscle development and muscle regeneration? If so, how it is regulated given that Alu are primate specific?

4. The effect of ADAR1 or hnRNP C knockdown is minimal. The authors explain this by the fact that the inclusion isoform is degraded by NMD. Nevertheless, knockdown of UPF1, which blocks NMD, causes a similar minimal effect. Moreover, even if the inclusion (PTC+) isoform is unstable, Figure 1D shows no obvious decrease in the skipping isoform (which must decrease when splicing is shifted toward inclusion). This again is in line with a marginal effect of ADAR1 and hnRNP C knockdown. The authors must hence address this issue. For example, they could measure the levels of the inclusion product following ADAR1 + UPF1 and hnRNP C + UPF1 double knockdown. Also, the way they use RT-qPCR to measure the isoforms (purification of the bands followed by qPCR) is cumbersome. Why did they not perform directly RT-qPCR with primers specific for the inclusion and specific for the skipping isoforms?

5. In Figure 2e, it is very surprising to see that MyoD is not expressed at D0 of differentiation given that MyoD is known to be expressed in both proliferating and differentiating muscle cells. Is it a strange behavior of the Hu5/KD3 cell line? Given that Hu5/KD3 are immortalized cells, it is important that the authors confirm results of Figure 2c and 2e (for selected targets) in primary human muscle cells from multiple donors to support the physiological relevance and generality of findings.

6. In reference 37, the pattern of SEPN1 protein expression is different compared to Figure 2c. First, in ref. 37 SEPN1 full-length is already strongly decreased at day1. Second, expression of full-length SEPN1 is back up at day 7. Is the discrepancy due to the use of immortalized (this study) instead of primary muscle cells (ref. 37)? Can the authors also document a rebound of SEPN1 skipping isoform at day 7 of differentiation? If so, how does it correlate with the levels of editing, ADAR1, hnRNP C, SEPN1 translation and NMD?

7. The results of Figure 2c do not agree with those of Figures 2e and S7. In Figure 2c, the total levels (full+short) of SEPN1 protein are grossly maintained during myogenic differentiation. Instead, Figures 2e and S7 show that the mature SEPN1 mRNA is almost absent after day 2 of differentiation. How can a protein be made if the RNA encoding it is almost absent? Is translation of SEPN1 increased during differentiation to compensate for very low mRNA levels? Is the short SEPN1 protein isoform much more stable than the long? Authors must address this discrepancy.

8. Figure S7 shows that the level of SEPN1 pre-mRNA is reduced by 50% by day 3 of differentiation. Hence, contrary to the authors claim, SEPN1 appears to be regulated also at the transcriptional level.

9. Are ROS or cytoplasmic/SR Ca²⁺ levels following SEPN1 levels during myogenic differentiation of Hu5/KD3 and primary human muscle cells?

10. The pattern of SEPN1 expression reported in the manuscript conflicts with the cited references reporting defects in terminally-differentiated, mature muscles from adult mice. If SEPN1 is normally not expressed in muscle fibers, how can muscles of adult SEPN1 KO mice display a phenotype? The authors must address this important discrepancy between in vitro and in vivo data.

11. The authors hypothesize that alteration of SEPN1 regulation causes a defect of myoblast fusion during differentiation leading to muscle atrophy. Nevertheless, they did not present any data supporting it. Actually, they did not provide any evidence that the complex mechanism of SEPN1 regulation that they describe has a biological relevance. Is myoblast fusion influenced by treatments affecting SEPN1 Alu exonization? Is ADAR1 or hnRNP C overexpression/knockdown affecting myoblast fusion? If so, can it be rescued/affected by overexpressing/knocking down SEPN1 full or short isoforms? Have the two SEPN1 protein isoforms different roles/requirements during myogenic differentiation? Is SEPN1 short required for myotube maturation? Most importantly, is Cas9-mediated prime editing of sequences needed for Alu exonization sufficient to affect SEPN1 regulation and the regulation of myoblast fusion? The mechanism described by the authors is very complex and it is unclear why it exists. If the goal is to downregulate SEPN1 levels during differentiation, there are simpler mechanisms that nature could have selected, like for example transcription downregulation, miRNA-mediated degradation or protein degradation. It is unclear why such a complex mechanism has been evolutionarily selected. Authors must provide data supporting the physiological role and the biological relevance of the mechanism they identified.

MINOR

A. Authors first cite Figure 4c in the Introduction. It would be more appropriate to have Figure 4c as Figure or Supplementary Figure 1.

B. The images of Figure 2B are of poor quality, making it difficult to evaluate differentiation.

C. The results on SepSecS, SEPHS1 and TRNAU1AP localization are shown on Figure 4f-h, not 3f-h.

D. The authors describe a number of results (Figures S14, S15, S16, S17, S18, Table S3) only in the Discussion section. This is very unusual. The results must be moved to the Results section and only their implications should be treated in the Discussion.

First, we thank the reviewers for their careful and helpful review of our manuscript, and for providing valuable suggestions for its improvement. Many critical points raised by the reviewers have provided us with the opportunity to collect additional data to address their concerns and to revise the manuscript accordingly. Our point-by-point responses to each of the reviewers' comments are shown below. Changes to the main text are marked in yellow.

Response to Reviewer #1's comments

The manuscript by Noda et al. identified post-transcriptional regulatory mechanisms of SEPN1 expression during myoblast-to-myotube differentiation. They demonstrated different regulatory mechanisms from early to late stages during myoblast differentiation. Overall, the experiments were well organized.

1. The authors demonstrated that A-to-I RNA editing is dynamically altered in AS-Alu1 of SEPN1 mRNA during myoblast differentiation as the steady-state level of ADAR1 gradual change, but there is no direct evidence to prove that those sites, especially mentioned site No. 46, are edited by ADAR1. It is necessary to analyze A-to-I RNA editing in AS-Alu1 of SEPN1 mRNA in Hu5/KD3 cells upon overexpression and knockdown of ADAR1.

Response: Because the transfection efficiency of Hu5/KD3 cells is quite low, we performed knockdown experiments targeting ADAR1 and ADAR2 in HeLa cells by transfecting them with siRNA. As shown in **Figure S6** (below), the editing frequencies at site Nos. 46, 57, and 60 in AS-Alu1 decreased upon ADAR1 knockdown, but did not change upon ADAR2 knockdown, demonstrating that these sites are edited by ADAR1, not by ADAR2.

Figure S6. ADAR1-mediated A-to-I RNA editing in AS-Alu1.

Frequencies of the editing site Nos. 46, 57, and 60 in AS-Alu1 of *SELENON* mRNAs were measured by RT-PCR of total RNA extracted from HeLa cells transfected with siRNAs targeting luciferase (control), ADAR1 and ADAR2 (Figure 1d, S3). The error bars represent the S.D.; n=3.

2. Sanger sequencing was used to evaluate the editing frequency at each site. The evaluation based on the peak areas by sequencing may not be accurate. I recommend the authors to carry out colony sequencing or next-generation sequencing to examine the editing frequency at each site.

Response: As suggested, we attempted to evaluate the editing frequency using NGS approaches. Because the A-to-I editing sites are present in Alu elements of intronic regions in pre-mRNA, we failed to obtain sufficient read depth by NGS to evaluate the editing frequency accurately. Previously, we showed that the A-to-I editing frequency correlates well with the A/G peak height ratio in the Sanger sequencing chromatogram (Supplementary Figure 7 in Sakurai et al., *Nature Chem Biol.*, 6, 733-740, 2010). We have cited this paper to explain the validity of this method.

3. In the middle stage, hnRNP C decreases. What's the reason?

Response: We were unable to determine the mechanism underlying regulation of hnRNP C during myogenesis. The fourth paragraph of the Discussion includes the statement that “the mechanism that regulates hnRNP C remains unknown”.

4. In the third paragraph of “Recoding regulation suppresses SEPNI expression in the late stage of myoblast Differentiation”, some Figures are not correctly marked. For example, Figure 3f, 3g, 3h should be changed to Figure 4f, 4g, 4h.

Response: These errors have been corrected.

5. In Figure 5e and S17, why is the retention time of the same substance different?

Response: Because substances were separated by capillary liquid chromatography, their retention times varied due to conditioning of the capillary column. The identity of the target molecules was therefore assessed by examining their accurate mass and product ions on CID.

6. In Figure 5e, the peaks of CCA-Sec-Alk+AC are not really like peaks. I recommend the authors to redo the experiments.

Response: The peaks were assigned according to their accurate mass, unique distribution of Se isotopes, and product ions of CID. As an example, the graphs below show mass spectra for CCA-Sec-Alk+AC on Days 0 and 5. The peak intensities (NL values) were sufficient to identify the target molecules.

• Day 0

• Day 5

Response to Reviewer #2's comments

Nota et al. show that an Alu-element in intron 2 of the human SELENON gene is exonized to some extent leading to NMD of the aberrant mRNA that includes a premature termination codon. The authors show that this exonization is suppressed by ADAR1-dependent A-to-I editing of the Alu-sequence and suggest that this occurs due to impaired recognition of the mRNA by U1 snRNA. Using a minigene they convincingly support this idea. HNRNP C competes with U1 snRNA and likewise suppresses exonization of the Alu sequence.

Then the authors use a human myoblast differentiation cell model and show that while SELENON pre mRNA levels are constant, SELENON mRNA levels decline in late stages of differentiation - suggesting that post-transcriptional processes account for this finding. In addition, the authors find that premature termination apparently occurs at the Sec codon and sensitivity to UPF1 knockdown suggests NMD as mechanism. The authors then go on and suggest that ADAR1 and HNRNP-C cooperate in early and mid stages of differentiation in Alu exon skipping, but their loss of expression in late myoblast differentiation accounts for the observed downregulation of SELENON mRNA and protein.

Up to there the story is flawless in the eyes of this reviewer, even if the authors address

the elephant in the room, i.e. why exonization is minimal in all other cell types and organs studied, only in the discussion and supplement S15. There we learn that muscle is the organ with lowest ADAR1 and HNRNP C expression.

Response: The introduction and the first section of the Results both mention that skeletal muscle is the organ with the highest frequency of Alu exonization of *SELENON* mRNA (Figure S2), and low level of expression of ADAR1 and hnRNP C in GTEx data (Figure S4).

Then the authors find by RNASeq that many factors previously associated or co-IP'ed with selenoprotein biosynthesis are downregulated in late myoblast differentiation. While in principle the reasoning is convincing, this reviewer challenges the view that all of these observed expression changes really influence SELENON translation.

Response: We have revised the related descriptions based on the suggestions made by this reviewer.

E.g. it is not so clear that SEPHS1 is "essential" for selenoprotein translation. While Tamura et al. suggested a function for Sps1 in their paper, doubts remained in the field. For example, targeted inactivation of *Sephs1* in mouse liver, a well-characterized model system of selenoprotein expression, did not reduce selenoprotein expression in general, but reduced *Gpx1* moderately and *Selenow* strongly. However, non-selenoproteins like *Glrx1* and *Gsto1* were massively reduced as well - questioning the specificity and the "essentiality" for selenoproteins (Tobe et al., 2016). If the high fold-change of SEPHS1 regulation in this manuscript is to be a convincing argument for its relevance for the mechanism in mind, then the high fold-change of *Glrx1* in the Tobe paper should be regarded equally telling - and then there the link to selenoproteins is not bigger than to *Glrx1* and *Gsto1*!

Response: Although SEPHS1 is an apparent homolog of bacterial SPS, it is unclear whether mammalian SEPHS1 is essential for selenoprotein biogenesis. We have added information about mouse *Sephs1* by citing Tobe et al., *Biochem J.*, 473, 2141-2154 (2016) in the subsection of the Results entitled "Recoding regulation suppresses SELENON expression...".

Similarly, in mouse liver, but also in neurons, *Trnaup1* inactivation (done using two

different genomic constructs) did not alter selenoprotein expression whatsoever, a finding hard to reconcile with an "essential" function (Mahdi et al., 2014).

Response: The same section also mentions the controversial role of mouse *TrnaU1ap*.

Finally, nucleolin (NCL) was initially identified as a SECIS-binding protein, but later dismissed, at least in the sense that it does not support Sec incorporation in the in vitro translation assay as does SECISBP2.

Response: As suggested, we have removed the related description of NCL.

Given that the observed effect of SELENON downregulation can likely be produced by downregulation of the two undoubtedly essential factors PSTK and SEPHS2 alone, it seems premature based on the data presented to promote SEPHS1 and TRNAU1AP, possibly also NCL, as "essential" selenoprotein biosynthesis factors.

Response: As suggested, we stressed that PSTK, SEPHS2, and SEPSECS were downregulated during the late stage of myoblast differentiation. The description of the regulation of the other factors has been toned down.

This reviewer, possibly with the majority in the field, rather considers them still in search for their real molecular function - not excluding physical interactions with selenoproteins.

SEPHS1 is called a Sec recycling factor in the manuscript, a function that seems to be occupied by Sec-lyase. At least, genetic inactivation of the Sec-lyase has the expected effect on selenoprotein expression in selenium deficiency (Byrns et al., 2014).

Response: We analyzed SCLY expression in the RNA-seq data and found that it did not change significantly during myoblast differentiation (Figure 4d).

Similarly, after fifteen years of published research, associations have been repeatedly found, but experimental support for the function of the different "super-complexes" that Small-Howard et al reported in 2006 are not known to this reviewer. If any of these complexes relied in their function on any of the above mentioned proteins, this should have become evident by now. Thus, to interpret "a slight reduction in the signals" of factors not essential to the process and maybe involved in "supercomplexes" whose

functions have not been ascertained, seems a bit stretched at the moment (Fig. 4C is well done, but a lot of speculation). This reviewer strongly suggests to stay with the solid data on editing and tRNA. If the authors want to endorse SEPHS1, TRNAU1AP, and nuclear "supercomplexes" whose functions remain elusive, they would need to show more convincing data on this. For the sake of this study on editing, these observations may not be necessary to show at all. If no compelling other data is available, the manuscript would profit massively, by simply cutting out SEPHS1 and TRNAU1AP. The data on SEPSECS is fine, convincing, and makes the point anyway.

The reduction of Sec-tRNA(Sec) in late differentiation is beautifully explained by reduction of SEPSECS (because the precursor, pSer-tRNA(Sec) is increased). Why SEPHS1 should have a hand in this - and, strangely, why SEPHS2 is not mentioned, remains unclear to this reviewer. In fact, if SEPHS1 would be limiting, how would the authors explain the increase of its downstream product pSer-tRNA(Sec)?

Response: As mentioned by this reviewer, downregulation of SEPSECS nicely explains Sec-tRNA(Sec) reduction and pSer-tRNA(Sec) accumulation. This may be the main reason for suppression of SELENON recoding during the late stage of myoblast differentiation. We also mentioned that SEPHS2 and PSTK mRNAs were slightly decreased during this late stage. Although we did not want to emphasize small changes in their expression, a plot of all factors involved in Sec-insertion in our RNA-seq data showed that both SEPHS1 and TRNAU1AP decreased significantly (Figure 4d). This has been mentioned briefly.

Discussion:

The authors may reconsider whether the details of T-tubules/Cav1.1 etc are needed or are simply elementary muscle physiology.

Response: We agree. The related description was removed.

What might be more relevant to the main message of this paper is the answer to the thought mentioned in the discussion: "Some pathogenic mutations in SEPN1 may cause dysregulation of Alu exonization." Which?

Response: The sentence in the sixth paragraph of the Discussion has been revised as follows:

"Future studies may identify pathogenic mutations that cause dysregulation of Alu

exonization in *SELENON* mRNA.”

The manuscript is well written, data are explained very clearly, experiments support the conclusions with the exceptions mentioned above. Clearly, this laboratory capitalizes on its technical championship with small RNA. Congratulations!

Response: We thank this reviewer for this statement!

Why not tell the selling point earlier rather than late in the discussion: "The SEPNI Alu exon is the first example of experimental proof of an mRNA in which A-to-I RNA editing at the U1 snRNA recognition site regulates mRNA splicing."?

Response: This sentence has been revised as follows:

“To the best of our knowledge, the *SELENON* Alu exon is the first experimental demonstration of an mRNA in which A-to-I RNA editing at the U1 snRNA recognition site regulates mRNA splicing.” Moreover, this is also mentioned in the subsection of the Results entitled “A-to-I editing at the 5’SS antagonizes ...”.

The reviewer strongly suggests to stringently adhere to the unified nomenclature on selenoproteins (Gladyshev et al., 2016). So SBP2 should read SECISBP2, SepSecS SEPSECS, SEPNI should be SELENON, Gpx1 GPX1, if referring to the protein, Gpx1, if referring to mouse mRNA etc.

Response: All nomenclature has been revised accordingly.

Response to Reviewer #3’s comments

In this manuscript the authors study the regulation of the selenoprotein SEPNI during myotube differentiation. The SEPNI pre-mRNA is modified by alternative splicing which itself is regulated by ADAR1 mediated editing and hnRNP C binding.

Evidence is provided that indicate that A to I editing at the U1 binding site will weaken the U1 interaction at the cryptic splice site.

Moreover, a short isoform of SEPNI occurs at the late stages of differentiation which was shown to be dependent on premature termination at the Selenocysteine codon. This process is controlled by downregulation of the selenocysteine tRNA which in turn is

controlled by the factors leading to tRNA-Sec formation.

The manuscript is well written and the claims are supported by the performed experiments. However, prior to publication several points should be clarified and addressed:

1) Abstract: The abstract does not mention the second part of the story, the premature termination as a result of lacking tRNA Sec. Also, the abstract mentions “In the middle stage” without mentioning “of myoblast differentiation”. Without the full statement, the sentence is misleading.

Response: We realize that the original abstract lacked critical information, and have revised it accordingly.

2) Page 6 first paragraph: “deficient” is spelled wrongly.

Response: This typo has been corrected.

3) Results page 8: the experiments are started with HeLa and then switch to Hek, most likely for the better protein production in Hek cells. This should be explained.

Response: Yes. This is correct. This description is provided in the subsection of the Methods section entitled “Vector construction and transfection”.

4) Figure 2c: addition of a molecular weight marker to the blot would be beneficial

Response: Molecular weight markers have been added to **Figure 2c**.

5) Page 12 top paragraph. A reference to the tight binding of hnRNP C to the Alu element intron should be given.

Response: This paragraph now cites the appropriate references: Zarnack et al., *Cell*, 152, 453-466 (2013) and König et al., *Nat Struct Mol Biol*, 17, 909-915 (2010).

6) Figure 4a) the schematic drawing of full length, shortened protein and suppressor codon protein is not correct (should be full length again).

Response: This error has been corrected.

7) Figure 4e, this blot needs quantification as the described differences are not obvious to the reader

Response: As suggested, the blot for SEPSECS was quantified. The blot for SEPHS1 has been removed, as suggested by Reviewer #2.

8) Figure 5c: why is the 3 day timepoint missing? What is the shorter band?

Response: According to western blot data for SEPNI (Figure 2c), the truncated product showed slight accumulation on Day 3, but greater accumulation on Day 5. We therefore isolated tRNA^{Sec} from Hu5/KD3 cells on Days 0 and 5. The shorter band, which was identified as tRNA^{Arg} by RNA-MS, is a contaminant probably resulting from non-specific hybridization to the DNA probe. This information is included in the figure legend.

Response to Reviewer #4's comments

Noda et al investigated the regulation of SEPNI during myogenic differentiation of the immortalized human muscle cell line Hu5/KD3. They describe a complex mechanism involving the sequential action/cooperation of alternative splicing, editing, translation and NMD. Intriguingly, data are presented that those similar mechanisms might regulate other selenoproteins.

While the data are novel and interesting, there are a number of issues that need to be addressed as listed below.

MAJOR

1. The authors did not provide any justification for why they decided to look first at the role of Alu exonization in SEPNI regulation. Why did they not consider simpler/more obvious mechanisms like RNA transcription or protein translation? See also point 8.

Response: *SEPNI* mRNA is a unique transcript bearing an Alu exon in its second intron, and previous studies show that the level of the Alu exon is quite high in skeletal muscle (Figure S2). In addition, we found A-to-I editing in the intronic Alu elements in *SEPNI* pre-mRNA. We therefore decided to look first at the post-transcriptional regulation of *SEPNI* expression. The Introduction includes additional background information about the Alu exon of *SEPNI* mRNA, explaining our justification.

2. For their work, the authors used cell lines like HeLa and Hu5/KD3.

To support the validity of their claims, the authors must show that the regulation of *SEPNI* during myogenic differentiation and the mechanisms they propose are conserved in primary muscle cells from multiple human donors.

Response: Although we agree that this regulation should be assessed in primary muscle cells to confirm our observations, it is difficult to ensure reproducibility of these cell preparations. It is known that human primary myogenic cells show limited proliferation capacity and progressive alteration of their characteristics during passage. The limitation for usage of human primary myogenic cells does not allow us to carry out both qualitative and quantitative analyses with high reproducibility. This is the reason for many researchers in this field to use immortalized myoblasts. To date, most studies on gene expression and regulation during myoblast differentiation have been performed using immortalized myoblasts such as the mouse C2C12 cell line. However, mouse myogenic cells do not always use the same pathways to control proliferation and differentiation as human myogenic cells. Hu5/KD3 cells are derived from Hu5 cells, which are human myogenic progenitor cells from a healthy individual. Previous studies confirm that Hu5/KD3 cells preserve phenotypic characteristics of the primary myogenic cells, retain the differentiation and proliferation abilities *in vitro*, and contributes to muscle differentiation upon xenotransplantation to immunodeficient mice under conditions of regeneration following muscle injury. In addition, Hu5/KD3 cells do not show anchorage-independent growth *in vitro* and tumor formation *in vivo* (Shiomi et al., *Gene Ther.*, 18(9):857-866, 2011). Moreover, Hu5/KD3 cells were used for measuring muscle contractility (Nagashima et al., *Adv Biosyst.*, 4(11):e2000121, 2020), single-nucleus RNA-seq during myoblast differentiation (Zeng et al., *Nucleic Acids Res.*, 44(21): e158, 2016) and the study of spinal and bulbar muscular atrophy (Iida et al., *Nat Commun.*, 10: 4262, 2019). These studies support the reliability of Hu5/KD3 cell as a differentiation model system of human skeletal muscle.

3. Given that the authors claim that the control of *SEPN1* expression they elucidated is important for the regulation of myoblast fusion and muscle contraction, it must be evolutionarily conserved. Is mouse *SEPN1* regulated during in vitro myogenic differentiation, and in vivo during muscle development and muscle regeneration? If so, how it is regulated given that Alu are primate specific?

Response: This is a very important comment. The Alu element is a primate specific SINE, and huge numbers of Alu elements are present in intronic regions of human genes. Sense and antisense strands of Alu elements form a long dsRNA in human pre-mRNAs. The vast majority of A-to-I RNA editing sites in the human transcriptome are present in dsRNA regions formed by inverted Alu elements. Moreover, A-to-I RNA editing is abundant in primates, but rare in mice, with human transcriptomes having 30 times more A-to-I RNA editing sites than mouse transcriptomes. In addition, Alu elements often provide cryptic splicing signals, with parts incorporated during splicing. Based on these findings, human-specific post-transcriptional regulation was likely mediated by Alu element and A-to-I RNA editing.

According to the study by Castets et al., *BMC Dev Biol.*, 9, 46 (2009), mouse *SEPN1* is also downregulated in C2C12 cells during myoblast differentiation, suggesting that *SEPN1* regulation is important physiologically. However, the absence of Alu from mice indicates that regulatory mechanisms may differ in mice and humans.

4. The effect of ADAR1 or hnRNP C knockdown is minimal. The authors explain this by the fact that the inclusion isoform is degraded by NMD. Nevertheless, knockdown of UPF1, which blocks NMD, causes a similar minimal effect. Moreover, even if the inclusion (PTC+) isoform is unstable, Figure 1D shows no obvious decrease in the skipping isoform (which must decrease when splicing is shifted toward inclusion). This again is in line with a marginal effect of ADAR1 and hnRNP C knockdown. The authors must hence address this issue. For example, they could measure the levels of the inclusion product following ADAR1 + UPF1 and hnRNP C + UPF1 double knockdown.

Response: We realized that single knockdown had slight effects on the Alu exonization of *SEPN1* mRNA. We therefore performed double knockdown experiments (Figure 1e). Simultaneous knock down of ADAR1 and UPF1 increased the fraction of the inclusion isoform from 4.6% to 14%, whereas simultaneous knock down of both hnRNP C and

UPF1 increased the fraction of the inclusion isoform from 6.2% to 19.3%. These findings indicate that the inclusion isoform generated by Alu exonization is constantly degraded via the NMD mechanism, and that the level of the skipping isoform is downregulated significantly. Because ADAR1 and hnRNP C redundantly block Alu exonization, the level of the inclusion isoform would likely be further increased when ADAR1 and hnRNP C are silenced simultaneously with UPF1 knockdown. Although we performed these triple knockdown experiments, the knockdown efficiency was insufficient for measuring the Alu exonization of *SEPNI* mRNA. Given that both ADAR1 and hnRNP C are lowly expressed in human skeletal muscle (Figure S4), a large fraction of *SEPNI* mRNA should be degraded by NMD.

Also, the way they use RT-qPCR to measure the isoforms (purification of the bands followed by qPCR) is cumbersome. Why did they not perform directly RT-qPCR with primers specific for the inclusion and specific for the skipping isoforms?

Response: We purified PCR product for each isoform to construct a standard curve for absolute quantification; this is because the PCR amplification efficiency of each isoform is different. For RT-qPCR, we used a set of primers specific for each isoform. We changed the description of the Methods section “RT-PCR”.

5. In Figure 2e, it is very surprising to see that MyoD is not expressed at D0 of differentiation given that MyoD is known to be expressed in both proliferating and differentiating muscle cells. Is it a strange behavior of the Hu5/KD3 cell line? Given that Hu5/KD3 are immortalized cells, it is important that the authors confirm results of Figure 2c and 2e (for selected targets) in primary human muscle cells from multiple donors to support the physiological relevance and generality of findings.

Response: As suggested, the amount of MyoD mRNA is relatively low on Day 0 before differentiation, as shown in the heat map in **Figure 2e**. This heat map, however, shows the relative levels of target mRNAs. In our RNA-seq data on Day 0, FPM value of MyoD mRNA is 83.3 [$\log_2(\text{FPM})=6.38$]. As shown in the violin plot (see right panel) for distribution of \log_2 (FPM) of all transcripts expressed on Day 0, the value 6.38 ranks relatively high level in the plot, indicating that MyoD mRNA is surely expressed on Day 0 in Hu5/KD3 cells. Supporting this observation, a study using mouse C2C12 cells (Panda et al., *Mol Cell Biol*, 34, 3106-3119, 2014) found that the steady-state level of MyoD mRNA is significantly lower before than during differentiation.

6. In reference 37, the pattern of SEPNI protein expression is different compared to Figure 2c. First, in ref. 37 SEPNI full-length is already strongly decreased at day1. Second, expression of full-length SEPNI is back up at day 7. Is the discrepancy due to the use of immortalized (this study) instead of primary muscle cells (ref. 37)? Can the authors also document a rebound of SEPNI skipping isoform at day 7 of differentiation? If so, how does it correlate with the levels of editing, ADAR1, hnRNP C, SEPNI translation and NMD?

Response: Figure 7 from reference 37 (right panel) shows detection of SEPNI expression by western blotting (WB) during myoblast differentiation. Lane 1 is a control fibroblast data. Lane 2 shows proliferating myoblasts, and lanes 3, 4, 5, and 6 correspond to cells 1, 3, 5 and 7 days, respectively, after differentiation. When the bands in lanes 2 and 3 (red boxes) were normalized relative to desmin, SEPNI was lower on Day 1. Emerin, the other control, was also similarly reduced, indicating that the level of expression of SEPNI relative to that of control proteins depends on these controls.

1052 *Human Molecular Genetics*, 2003, Vol. 12, No. 9

Figure 7. Expression of SEPNI during myoblast differentiation. Western blot analysis was performed on proteins extracted from fibroblasts (lane 1), proliferating myoblasts (lane 2) and myoblasts cultivated in a fusion medium for 1, 3, 5 or 7 days (lanes 3-6, respectively). Anti-desmin antibodies were used as a specific marker of muscular cells, since this protein is expressed both by the proliferating myoblasts and the differentiated myotubes.

RNA-seq data (right panels) showed that desmin expression increased, whereas emerin expression decreased, from Day 0 to Day 1. Thus, if the western blotting data were normalized to desmin, SEPN1 did not decrease on Day 1. Because we used GAPDH as a loading control (Figure 2c), we observed a different pattern of SEPN1 expression. In addition, we could not quantify the exact amount of target protein by western blotting in the absence of a standard curve.

Regarding rebound expression of SEPN1 on Day 7 (blue box in Figure 7), we did not perform any experiments with Hu5/KD3 cells on Day 7. To synchronize myoblast differentiation, however, we eliminated proliferating myoblasts by treatment with cytarabine. Because SEPN1 is produced by proliferating myoblasts, we did not observe a clear reduction of SEPN1 during differentiation without cytarabine treatment. The rebound SEPN1 expression on Day 7 observed in the previous study may have been due to the proliferating myoblasts that remain among differentiated cells.

7. The results of Figure 2c do not agree with those of Figures 2e and S7. In Figure 2c, the total levels (full+short) of SEPN1 protein are grossly maintained during myogenic differentiation. Instead, Figures 2e and S7 show that the mature SEPN1 mRNA is almost absent after day 2 of differentiation. How can a protein be made if the RNA encoding it is almost absent? Is translation of SEPN1 increased during differentiation to compensate for very low mRNA levels? Is the short SEPN1 protein isoform much more stable than the long? Authors must address this discrepancy.

Response: Figure 2e shows that the steady-state level of *SEPN1* mRNA decreased on Day 3 of differentiation. The heat map presentation, however, shows the relative level of target mRNA. *SEPN1* mRNA is not absent, but is actually expressed during the middle and late stages of differentiation. RT-qPCR showed that, compared with Day 0, mature *SEPN1* mRNA is reduced to about 20–30% on Days 3–5 (Figure S11). Because the steady-state level of protein does not always correlate with mRNA level, the stability of two isoforms may differ.

8. Figure S7 shows that the level of SEPN1 pre-mRNA is reduced by 50% by day 3 of differentiation. Hence, contrary to the authors claim, SEPN1 appears to be regulated also at the transcriptional level.

Response: We agree that *SEPN1* mRNA is transcriptionally regulated during myoblast differentiation, as well as being regulated at the post-transcriptional level. We have

added a relevant description of this point to the subsection of the Results entitled, “Suppression of Alu exon maintains *SELENON* mRNA integrity...”.

9. Are ROS or cytoplasmic/SR Ca²⁺ levels following SEPNI levels during myogenic differentiation of Hu5/KD3 and primary human muscle cells?

Response: We did not measure ROS and Ca²⁺ levels in Hu5/KD3 cells directly during differentiation. However, RNA-seq data showed that ER stress markers were transiently elevated during the middle stage of differentiation (Figure 2e). ER stress induces the UPR pathway, leading to activation of ERO1, which in turn generates ROS. Because Hu5/KD3 cells retain the same differentiation and proliferation abilities as human myogenic progenitor cells, the same signaling pathways and mechanism(s) should be involved in myoblast cell fusion mediated by ER Ca²⁺ depletion, ER stress, UPR, and ROS production during myoblast differentiation of Hu5/KD3 cells.

10. The pattern of SEPNI expression reported in the manuscript conflicts with the cited references reporting defects in terminally-differentiated, mature muscles from adult mice. If SEPNI is normally not expressed in muscle fibers, how can muscles of adult SEPNI KO mice display a phenotype? The authors must address this important discrepancy between *in vitro* and *in vivo* data.

Response: We cannot compare mouse *in vivo* data directly with human *in vitro* data. *Sepn1* is highly expressed during mice embryogenesis, especially in the myotome. However, no defects in muscle development and growth were detected in *Sepn1* knock-out mice, suggesting that other redox proteins might compensate for *Sepn1* function. By contrast, *SEPNI* is essential for muscle regeneration and satellite cell maintenance in mice and humans. These findings might partly explain the molecular pathogenesis of *SEPNI*-related myopathies. Further studies are necessary to determine the physiological activity of SEPNI function and its regulation during myoblast differentiation.

11. The authors hypothesize that alteration of SEPNI regulation causes a defect of myoblast fusion during differentiation leading to muscle atrophy. Nevertheless, they did not present any data supporting it. Actually, they did not provide any evidence that the complex mechanism of SEPNI regulation that they describe has a biological relevance. Is myoblast fusion influenced by treatments affecting SEPNI Alu exonization? Is ADAR1 of hnRNP C overexpression/knockdown affecting myoblast fusion?

If so, can it be rescued/affected by overexpressing/knocking down SEPNI full or short isoforms?

Response: As suggested by this reviewer, we are eager to perturbate the regulatory mechanism of SEPNI expression and observe its effect on myoblast differentiation. Although we attempted to introduce siRNAs and plasmids into Hu5/KD3 cells using various transfectants and conditions, the transfection efficiency of this cell line is quite low, and we were unable to find any practical methods to manipulate this cell line. Additional studies are necessary.

Have the two SEPNI protein isoforms different roles/requirements during myogenic differentiation? Is SEPNI short required for myotube maturation?

Response: The short isoform is a truncated SEPNI bearing TM and EF-hand, but lacking a catalytic domain. According to recent study (Filipe et al., *Cell Death Differ*, 28, 123–138, 2021), ER Ca²⁺ uptake of *SEPNI*-deficient cells was not rescued by SEPNI mutants, in which the Sec recoding site was replaced by Cys or Ser, demonstrating that an active site with Sec is necessary for SEPNI function. Thus, the short isoform should not have the catalytic activity of SEPNI. However, the short isoform is quite stable during myoblast differentiation, indicating that it might have some biological function. We have added relevant descriptions about the functional aspects of the short isoform to the eighth paragraph of the Discussion section.

Most importantly, is Cas9-mediated prime editing of sequences needed for Alu exonization sufficient to affect SEPNI regulation and the regulation of myoblast fusion? The mechanism described by the authors is very complex and it is unclear why it exists. If the goal is to downregulate SEPNI levels during differentiation, there are simpler mechanism that nature could have selected, like for example transcription downregulation, miRNA-mediated degradation or protein degradation. It is unclear why such a complex mechanism has been evolutionarily selected. Authors must provide data supporting the physiological role and the biological relevance of the mechanism they identified.

Response: We were also puzzled by the complex mechanism required by SEPNI to regulate its expression. One simple answer is that SEPNI is a selenoprotein that is

synthesized by the UGA/Sec recoding system. *SEPNI* expression is intricately intertwined with NMD, alternative splicing of the Alu exon, and A-to-I RNA editing. Assessing the physiological role of this mechanism requires manipulation of Hu5/KD3 cells. However, as described above, we were unable to find any practical method for transfection of this cell line. Future studies are needed to determine the biological importance of this mechanism.

MINOR

A. Authors first cite Figure 4c in the Introduction. It would be more appropriate to have Figure 4c as Figure or Supplementary Figure 1.

Response: This figure is now Figure S1.

B. The images of Figure 2B are of poor quality, making it difficult to evaluate differentiation.

Response: As suggested, the images in Figure 2B have been replaced by better images.

C. The results on SepSecS, SEPHS1 and TRNAU1AP localization are shown on Figure 4f-h, not 3f-h.

Response: These typos have been corrected.

D. The authors describe a number of results (Figures S14, S15, S16, S17, S18, Table S3) only in the Discussion section. This is very unusual. The results must be moved to the Results section and only their implications should be treated in the Discussion.

Response: The manuscript has been revised to include these results in the Results section.

REVIEWERS' COMMENTS

Reviewer #1 (Remarks to the Author):

The authors addressed my concerns. I don't have further questions.

Reviewer #2 (Remarks to the Author):

The authors have addressed all of this reviewer's concerns and made appropriate changes to this nice manuscript. Congratulations!

Reviewer #3 (Remarks to the Author):

In this revised version of their manuscript, the authors have addressed all concerns raised by me. Some of the points of critique overlapped with those of reviewer 2 and were addressed by responding to this reviewer.

Overall, the paper is reads easier and the data is presented more clearly.

I have no further points that need to be addressed.

Reviewer #4 (Remarks to the Author):

MAJOR POINT 2. The author's answer is unsatisfactory. The purpose of performing experiments on primary cells of multiple donors is to ensure that they are reproducible and robust enough to withstand interindividual variability. If the authors data are only reproducible in a single immortalized cell line, there is concern that they do not have relevance to real word regulation of myogenesis.

MAJOR POINT 9. The author's reply is unsatisfactory. Why not measure ROS or cytoplasmic/SR Ca² levels to determine if they are significantly correlated to SEP^N1 levels during myogenic differentiation? As such, the manuscript does not contain any functional characterization to support the relevance of the mechanism they described.

MAJOR POINT 11. The answer is unsatisfactory. Many groups (including us) perform knockdown routinely with high efficiency in primary and immortalized human muscle cells by siRNA transfection with lipofectamine. Similarly, overexpression can be done with high efficiency using lentiviruses. If Hu5/KD3 are not amenable to knockdown/overexpression, it is another reason to include additional human muscle cells in the study to support reproducibility and support the functional relevance of the complex pathway described by the author.

Response to Reviewer #1's comments

The authors addressed my concerns. I don't have further questions.

Response: We really appreciate the constructive and valuable suggestions to improve our manuscript.

Response to Reviewer #2's comments

The authors have addressed all of this reviewer's concerns and made appropriate changes to this nice manuscript. Congratulations!

Response: We really appreciate the constructive and valuable suggestions to improve our manuscript.

Response to Reviewer #3's comments

In this revised version of their manuscript, the authors have addressed all concerns raised by me. Some of the points of critique overlapped with those of reviewer 2 and were addressed by responding to this reviewer. Overall, the paper is reads easier and the data is presented more clearly.

I have no further points that need to be addressed.

Response: We really appreciate the constructive and valuable suggestions to improve this manuscript.

Response to Reviewer #4's comments

MAJOR POINT 2. The author's answer is unsatisfactory. The purpose of performing experiments on primary cells of multiple donors is to ensure that they are reproducible and robust enough to withstand interindividual variability. If the authors data are only reproducible in a single immortalized cell line, there is concern that they do not have relevance to real word regulation of myogenesis.

Response: Thank you very much for the suggestion. We understand this is an important point. However, if we plan to carry out the suggested experiments, we need to apply for clinical studies to our ethical and clinical research board. It will take long time to get approval. Please allow us to carry out such experiments in our future studies. We toned down the claims on physiological implications by removing "physiologically" from the following sentence in Discussion. "The regulation of Alu exonization of SELENON mRNA revealed in this study is physiologically important for maintaining the expression level of SELENON from the early-to-middle stage of myoblast differentiation."

MAJOR POINT 9. The author's reply is unsatisfactory. Why not measure ROS or cytoplasmic/SR Ca² levels to determine if they are significantly correlated to SEPN1 levels during myogenic differentiation? As such, the manuscript does not contain any functional characterization to support the relevance of the mechanism they described.

Response: Thank you very much for the suggestion. We agree that these experiments strengthen this manuscript. We would like to carry out these experiments in our future studies. In this manuscript, we removed "ER stress level" in the Figure 6.

MAJOR POINT 11. The answer is unsatisfactory. Many groups (including us) perform knockdown routinely with high efficiency in primary and immortalized human muscle cells by siRNA transfection with lipofectamine. Similarly, overexpression can be done with high efficiency using lentiviruses. If Hu5/KD3 are not amenable to knockdown/overexpression, it is another reason to include additional human muscle cells in the study to support reproducibility and support the functional relevance of the complex pathway described by the author.

Response: Thank you very much for the suggestion. In fact, we tried a number of time to transfect siRNAs and plasmids into Hu5/KD3 cells using various transfectants or lentiviruses. However, we could not achieve the clear effect unfortunately. We are eager to perturbate the regulatory mechanism of SELENON expression and observe what happens in the myoblast differentiation. We would like to perform overexpression/knockdown of ADAR1 and hnRNP C and also deletion of the Alu element with CRISPR/Cas9 by using primary cells of multiple donors in the future study. In this manuscript, we toned down the claims on physiological implications.